# Quantifying Global $N_2O$ Emissions from Natural Ecosystem Soils Using Trait-Based Biogeochemistry Models

Tong Yu[1] and Qianlai Zhuang[1,2]

[1]Earth, Atmospheric, and Planetary Sciences, Purdue University, West Lafayette IN 47907, USA

[2]Department of Agronomy, Purdue University, West Lafayette, IN 47907, USA

*Correspondence to*: Qianlai Zhuang (qzhuang@purdue.edu)

**Abstract** A group of soil microbes plays an important role in nitrogen cycling and $N_2O$ emissions from natural ecosystem soils. We developed a trait-based biogeochemical model based on an extant process-based

biogeochemistry model, the Terrestrial Ecosystem Model (TEM), by incorporating the detailed microbial physiological processes of nitrification. The effect of ammonia-oxidizing archaea (AOA), ammonia-oxidizing bacteria (AOB) and nitrite-oxidizing bacteria (NOB) was considered in modeling nitrification. The microbial traits including microbial biomass and density were explicitly considered. In addition, nitrogen cycling was coupled with carbon dynamics based on stoichiometry theory between carbon and nitrogen. The model was parameterized using

observational data and then applied to quantifying global $N_2O$ emissions from global terrestrial ecosystem soils from 1990 to 2000. Our estimates of $8.7 \pm 1.6$ Tg N $yr^{-1}$ generally agreed with previous estimates during the study period. Tropical forests are a major emitter, accounting for 42% of the global emissions. The model was more sensitive to temperature and precipitation, and less sensitive to soil organic carbon and nitrogen contents. Compared to the model without considering the detailed microbial activities, the new model shows more variations in response to

seasonal changes in climate. Our study suggests that further information on microbial diversity and eco-physiology features is needed. The more specific guilds and their traits shall be considered in future soil $N_2O$ emission quantifications.

## 1.      Introduction

Nitrogen (N) is the most abundant element in the atmosphere. It accounts for 78% of the Earth's atmosphere. NOx (referring to NO and $NO_2$) is a main pollutant in the air, especially in heavily populated areas. $N_2O$, a potent greenhouse gas, is also an important oxidizer in chain reactions in the air. Additionally, N is also an important nutrient for almost all living things. For plants and most microbes, N is not only the structural element to build their body, but also a fundamental element for enzyme involving in almost all metabolic processes. Chemical compounds of nitrogen encompass many oxide states ranging from -3 (ammonia) to +5 ($N_2O_5$). The cycle of nitrogen can thus be characterized by processes of oxidation and reduction, which is different from other element cycles such as sulfur (S) and phosphorous (P).

Microbial activity plays a crucial part in the Earth's biogeochemical cycles, affecting biological fluxes of H, C, N, O, and S (Falkowski et al., 2008). In the air and soils, the compounds of N exist in multiple oxidation states, but most of them are in oxidized states. When N is released from organism cells, it will be oxidized into other forms rapidly. The processes of nitrification and denitrification play an important role in this flow path. These biochemical reactions are highly related to micro-organisms. In the process of nitrification ($NH_3/NH_4^+ \rightarrow NO_2^-$ $\rightarrow NO_3^-$), ammonia-oxidizing bacteria (AOB) and nitrite-oxidizing bacteria (NOB) are the main metabolic labors (Prosser and Nicol, 2008). Nitrification in aerobic oxidation condition was first discovered in 1890 (Winogradsky, 1890), and it is a classical theory for the microbial activities till now. Though in recent years, anaerobic ammonia oxidation has been found in natural ecosystems (Francis et al., 2007), the aerobic oxidations by microbes especially by archaea and bacteria are still a dominant process in most circumstances.  In the first step, ammonia ($NH_3$) is changed to hydroxylamine ($NH_2OH$), and then is dissociated to $NO_2^-$ and water. This step requires aerobic conditions because $O_2$ acts as the terminal electron acceptor and ammonia acts as the electron donor. This is the rate-limiting step of nitrification. Beta- and gamma-proteobacteria (Kowalchuk and Stephen, 2001) and thaumarchaea (Brochier-Armanet et al., 2008) are responsible for this step. This reaction is catalyzed by chemolitho-autotrophic bacteria and archaea. The second step is from $NO_2^-$ to $NO_3^-$, which is conducted by nitrite oxidizing bacteria (NOB) belonging to five genera (Nitrobacter, Nitrospira, Nitrococcus, Nitrospina and Nitrotoga). Compared to the first step, it needs less energy. When $NO_2^-$ is produced in the first step, it gets oxidized in the second step almost instantly. Thus, it is unlikely for $NO_2^-$ to get accumulated in the soil. There are three groups of autotrophic AOBs. Two of them are β (Nitrosospira) and γ (Nitrosococcus) subclasses of the Proteobacteria, and the left one is within the

Planctomycetales (Kowalchuk and Stephen, 2001). In terrestrial environment, the population of AOBs is highly impacted by soil moisture, pH, nitrogen input and vegetation. If the soil is polluted, the population will also be profoundly affected. The gene of 16S rRNA sequence determines the ammonia oxidation for AOBs.

Archaea is critically important in the first step of nitrification, which is also one of the most widely distributed microorganisms on the Earth. The total amount of this microbe is in a magnitude of $10^{28}$ cells. The total cell in a human body is about $3.72 \times 10^{13}$ (Bianconi et al., 2013), so $10^{28}$ is even far more than the total cell number of all human beings in total on the Earth. The dominant gene related to nitrification is ammonia mono-oxygenase (amoA) according to the study in sea (Venter et al., 2004) and soils (Treusch et al., 2005). Compared to bacteria, which have only a small number of species related to nitrification, there are hundreds of amoA sequences involved in ammonia oxidation. Ammonia oxidized archaea (AOA) can be adapted to more habitats and environments, even including some suboxic zone (Francis et al., 2005). AOA is much more abundant than AOB (Leininger, et al, 2006). These organisms are dominant ammonia oxidizers both in soils and the sea and the activities of these archaea shall be represented in N biogeochemistry models.

Denitrification is a major source of nitric and nitrous oxide emissions into the atmosphere. This process includes several reductive processes and each reaction is performed by a wide range of microorganisms. In denitrification, nitrate is used as the terminal electron acceptor instead of $O_2$. For some bacteria, $NO_2^-$, $N_2O$ and $NO$ are the terminal electron acceptor. Compared to nitrification, there are more steps in denitrification ($NO_3^- \rightarrow NO_2^- \rightarrow NO \rightarrow N_2O \rightarrow N_2$). Although the final products are $N_2$, $NO$ and $N_2O$ as gases, which means they can escape during the process. If they are dissolved in soils, they will be utilized for the next step of reaction. Primarily, denitrification is catalyzed by bacteria (Torregrosa-Crespo et al., 2016) and archaea (Cabello, Roldán, & Moreno-Vivián, 2004), but some fungi (Fusarium oxysporum) can denitrify as well (Shoun et al.,2012). Denitrifying organisms also belong to bacteria and Archaea. Different species are responsible for certain steps for denitrification. Nitrite reductase (nirK and nirS genes) conducts the reaction from $NO_2^-$ to $NO$ (Priemé et al., 2002). Nitrous oxide reductase (nosZ gene) finishes the last step of denitrificatrion (Kandeler et al., 2006). Generally, more steps and more microbes are involved in denitrification than in nitrification. This study presented a trait-based model to assess some of these microbial activities that determine the nitrifying processes, particularly the limitation of nutrient supplies. The model describes the metabolisms and reproduction of nitrifying microbes, and their controls under environmental and soil conditions. Numerical simulations of $N_2O$ emissions from 1990 to 2000 were performed on both site and global

levels. Using the model, our research goals are to examine: (1) whether the detailed soil microbial traits would improve estimating soil emissions of $N_2O$ and (2) the role of carbon and nitrogen stoichiometry in nitrification. By using $N_2O$ flux data from 80 observational sites, we first calibrate and verify the model. The model is then used to analyze the pattern and seasonal variation of global $N_2O$ emissions from natural ecosystem soils from tropical to
polar areas.

## 2.    Method

### 2.1    Overview

We first revised the core carbon and nitrogen dynamics of TEM (Zhuang et al., 2003) by including more detailed N cycling and microbial dynamics effects (Figure 1). Second, the key parameters in the model were
calibrated using site-level observational data for global major vegetation types. Third, the model was tested based on data from 80 observational sites. Finally, the regional and global $N_2O$ emissions were estimated with the model for the last decade of the 20[th] century. In addition, the model sensitivity to various climate and soil conditions was tested.

### 2.2    Model Modification

We revised the terrestrial ecosystem model (TEM, Zhuang et al., 2003) to improve the core carbon and nitrogen dynamic module by incorporating the detailed nitrification process at a daily time step. The major processes of nitrogen dynamic module are inherited from Yu (2016), including the effect of physical conditions on both nitrification and denitrification, and the principles of stoichiometry of carbon and nitrogen dynamics in soils. Details
and equations describing nitrification, denitrification and $N_2O$ fluxes can be found in Yu (2016). The model in this study was further incorporated with the effects of the activity and biomass of nitrifier guilds on nitrification (Bouskill et al. 2012). In addition to the losses from oxidation, the N uptake by microbial biomass and the biomass breakdown by detoxification process were also modeled. The dynamics of ammonia concentration in soils are simulated as:

$$\frac{d[NH_3]}{dt} = -V_{Ox}^{NH_3} - \left(V_{AOO}^{NH_3} + V_{NOB}^{NH_3}\right) + \frac{1}{4}\left(D_o^{NO_2} + D_o^{NO}\right) \tag{1}$$

Where $\left[NH_3\right]$ represents the concentration of soil ammonia, including $NH_3$ and $NH_4^+$. $V_{Ox}^{NH_3}$ is the rate of ammonia oxidized by nitrifiers, calculated with the methods described in TEM (Yu, 2016). $V_{AOO}^{NH_3}$ and $V_{NOB}^{NH_3}$ are ammonia taken up by AOO and NOB, respectively, to support metabolism and reproduction of microbes. The last term of Eq.1 is the part consumed in detoxification process, and the reactions are described in Eq.8. The constant here represents the stoichiometry in detoxification reactions (Bouskill et al., 2012):

$$\frac{d[NO_2]}{dt} = V_{Ox}^{NH_3} - V_{Ox}^{NO} - D_{AOO}^{NO_2} \tag{2}$$

Where $\left[NO_2\right]$ represents the concentration of $NO_2$. $V_{Ox}^{NO}$ is the oxidization rate by NOB and $D_{AOO}^{NO_2}$ is the loss in the detoxification.

The consumption rate of $NH_3$ by AOA and AOB is determined by the concentration of $NH_3$ and $O_2$ in the soil. For the simulation of ammonia oxidation by ammonia-oxidizing organism, the cell biomass was considered in the Briggs-Haldane kinetics calculation (Koper et al., 2010):

$$V_{AOO}^{NH_3} = V_{max}^{NH_3} \frac{[NH_3]}{K_{AOO}^{NH_3} + [NH_3]\left(\frac{1+[NH_3]}{K_{AOO}^{NH_3}}\right)} \frac{[O_2]}{K_M^{O_2} + [O_2]} B_{TA} \tag{3}$$

Where $V_{max}^{NH_3}$ is the maximum substrate uptake rate for ammonia (M day$^{-1}$). This value varies between different guilds of microbes. The average value for AOB is about 0.5 and the average value for AOA is about 0.6. $K_{AOO}^{NH_3}$ is the half saturation constant for $NH_3$ (μM) and $K_M^{O_2}$ is the Michaelis-Menten parameter for oxygen (μM) (Table 1). $B_{TA}$ is the total cell biomass for ammonia oxidizing organisms (AOA+AOB).

The consumption of $NO_2^-$ is similar to Eq.3:

$$V_{NOB}^{NO_2} = V_{max}^{NO_2} \frac{[NO_2]}{K_M^{NO_2} + [NO_2]} \frac{[O_2]}{K_M^{O_2} + [O_2]} B_{TN} \tag{4}$$

Where, $K_M^{NO_2}$ is the maximum substrate uptake rate for $NO_2^-$ (M day$^{-1}$). This value also depends on different guilds, and the value could be from 0.4 to 4 (Bouskill et al., 2012); here 2.0 was used. $K_M^{NO_2}$ is the half saturation constant for $NH_3$ (μM) and $K_M^{O_2}$ is the Michaelis-Menten parameter for oxygen (μM). $B_{TN}$ represents the total cell biomass of NOB.

Considering the cell division of microbes, the growth of AOB biomass is (Bouskill et al., 2012):

$$\frac{dB_{TA}}{dt} = \mu_{max} min\{d_i\}B_{TA} - \varepsilon B_{TA} - \frac{1}{4}\left(D_A^{NO_2} + D_A^{NO}\right) \tag{5}$$

The first term $\mu_{max} min\{d_i\}B_{TA}$ is the cell division rate. $\mu_{max}$ (day$^{-1}$) is the nitrifier maximum specific

growth rate for ammonia oxidizing organisms (AOO). It is less than 0.1 for AOO, and here 0.05 is used. $min\{d_i\}$

represents the constraint of element. It is defined as the cell division of AOO or NOB, which is governed by Droop

kinetics (Droop, 1973):

$$d_B^i = max\left(1 - \frac{Q_B^{min}}{Q_B^i}, 0\right) \tag{6}$$

Q is the cellular quota for nitrogen or carbon. It is defined as $Q_N = B_N/B_T$, $Q_C = B_C/B_T$, which is the

percentage of a certain element in total biomass. For example, the cell division of N for a guild is:

$$d_{B,N}^1 = max\left(1 - \frac{1/13.2}{B_N/(B_N+B_C)}, 0\right) \tag{7}$$

According to the C: N ratio for nitrifiers, the amount of carbon is supposed to be 6.6 to 13.2 times of the amount of

N (Bouskill et al., 2012). If the ratio of C: N is greater than 1/13.2, the reproduction of microbe is limited by N. In

contrast, the process is limited by C if C: N is smaller than 6.6.

The second term $\varepsilon B_{TA}$ indicates the death rate. $\varepsilon$ is the mortality rate. The last term $\frac{1}{4}\left(D_A^{NO_2} + D_A^{NO}\right)$ refers

to the biomass loss for converting $NO_2$ to NO and NO to $N_2O$:

$$4NO_2 + CH_2O \rightarrow 4NO + CO_2 + 3H_2O$$

$$8NO + 2CH_2O \rightarrow 4N_2O + 2CO_2 + 2H_2O \tag{8}$$

Similarly, the growth of NOB biomass is (Bouskill et al., 2012):

$$\frac{dB_{TN}^i}{dt} = \mu_{max}^i min\{d_i\}B_{TN}^i - \varepsilon B_{TN}^i \tag{9}$$

The improved nitrogen dynamic module (NDM) explicitly simulates the effect of climate conditions on

nitrogen cycle, and the effects of detailed microbial activities were considered in nitrification and detoxification

processes. In addition, the processes of N deposition, mineralization, and denitrification were also modeled. The

influence of climate conditions and soil textures on the geochemical reaction conditions (e.g., soil temperature, pH,

and oxygen concentration) were also considered. The metabolism and reproduction of microbes, together with

several substrates (organic N, ammonia) determine the reaction rate. The soil thermal module (STM) and

hydrological module (HM) are inherited from TEM by Zhuang et al (2003). The NDM utilizes the soil temperature

simulated in STM and the soil water content is estimated with HM.

The values of parameters vary between different biomes and guilds. Based on literature review for the study of nitrifier guilds, the initial values for parameters are given in Table 1. Our study simulates AOO and NOB as individual guilds for each biome, and a uniform guild density is assumed across the biome.

### 2.3 Data

The $N_2O$ observational data from 1980 to 2010 for typical vegetation types were acquired from literature (Table 2). The observational sites are characterized by temperate coniferous forest, boreal forest, tundra, and succulent area. Annual site-level $N_2O$ emissions were collected, covering more than 10 biomes especially in temperate and tropical areas. The datasets were only from nonagricultural terrestrial ecosystems with experimental periods from several weeks to years. Four typical flux tower sites including tropical forests (1, dark green circles), grasslands (2, light green circles), temperate forests (3, yellow circles) and others (4, red circles) were selected to verify the modeled seasonal variation.

Global simulations were driven with spatially-explicit data of climate, soil conditions, vegetation types and land cover at a spatial resolution of 0.5º x 0.5º. Climate data including monthly cloudiness, precipitation, temperature and water vapor pressure are from Climate Research Unit (CRU). While the soil conditions, vegetation types and land cover types were assumed to be invariable over our study period, and only to vary over from grid to grid spatially. The details about global vegetation data and soil data were available in Zhuang et al (2003) and McGuire et al. (2001). Model runs were carried out at a daily step for the time period 1990-2000. The explicitly spatial data of soil water pH from ORDL gridded soil properties product (https://daac.ornl.gov/cgi-bin/dsviewer.pl?ds_id=546) are based on The World Inventory of Soil Emission Potentials (WISE) database (Batjes, 2000). There were two parts of nitrogen deposition data, including ammonia and nitrate. Wet deposition was estimated with rainfall nitrogen concentration from national trend network by the National Atmospheric Deposition Program (NADP) monitors, and precipitation data. Dry deposition data was collected from Aggregate Deposition data (1987-2016), by EPA's Clean Air Status and Trends Network (CASTNET). The global average carbon dioxide concentration observed at NOAA's Mauna Loa Observatory by parts per million was used uniformly (there is no spatially variation) as driving data.

The initial values of soil microbial carbon and nitrogen, and the ratio of C:N at the global scale were from a compilation of Global Soil Microbial Biomass Carbon, Nitrogen, and Phosphorus Data set (http://dx.doi.org/10.3334/ORNLDAAC/1264), compiled from comprehensive data survey of 315 publications from

11/16/1977 to 06/01/2012 (Xu et al, 2014). The microbial biomass data was collected mainly from the depth within 0-30 cm ($\mu$mol kg$^{-1}$), and compiled into two soil depths of 0-30cm and 0-100cm (g C m$^{-2}$ or g N m$^{-2}$), including carbon and nitrogen storage and C:N ratio for soil microbial biomass. The spatial data were converted from the original 0.05$^\circ$ x 0.5$^\circ$ to a resolution of 0.5$^\circ$ x 0.5$^\circ$, covering 12 biomes across the globe, which were consistent with our model simulation grids. The one-time estimate of spatially data was resampled to the spatial resolution of TEM. Twelve biomes in the dataset were boreal forest, temperate coniferous forest, temperate broadleaf forest, tropical/subtropical forest, mixed forest, grassland, shrub, tundra, desert, cropland and pasture.

**2.4 Model Calibration and Validation**

The model parameters related to N dynamics were calibrated at the site level for major representative ecosystems.  Parameter ranges and initial values were determined based on literature review (Table 1). Direct N$_2$O measurements for various terrestrial natural ecosystems including forests, grasslands, shrub lands and tundra, tropical and temperate areas where live more microbial species were organized (Table 2). All data were monthly average N$_2$O emissions measured with chambers and eddy flux techniques. The observations were conducted under different climate and soil conditions. The measurement periods covered from several days to several months and the time interval for measurement varied from seconds to days. If the time interval of emissions was less than one day, the emission values were calculated into monthly average. The meteorological conditions at the observation sites were retrieved from the original studies.  A quarter of the sites were used for calibration and the remaining were used for validation.

Parameterization was conducted only for natural terrestrial ecosystems. Parameters in Table 4 were adjusted individually while other parameters of model were kept as is. The parameters were optimized through altering parameters, iterating model simulations, and calculating the difference between observation and simulation. We apply the site-level parameters for representative ecosystem types to grid cells at 0.5$^\circ$ x 0.5$^\circ$ resolution at the global scale. The ecosystem types are listed in Table 2 and their distributions are from Melillo et al. (1993).

The field observational sites selected for model calibration and validation spread across major vegetation types and biomes (Figure.2). Eighty-one sets of observational data were collected from 60 publications, covering varieties of climate zone from semidry savanna to rainforest, polar to tropical areas.  26 sites were from tropical rainforests, 22 from temperate grassland and savanna, 21 from temperate forests and the rest from 9 other vegetation

types. The monthly or daily average temperature varied from -10.5 to 42 °C, with the precipitation from 0.1 to 3962 mm, representing diverse climate conditions.

### 2.5 Model Sensitivity

To test model sensitivity to forcing data, simulations at both site and regional levels were conducted. The monthly average air temperature (TAIR), precipitation (PREC), cloudiness (CLDS) and water vapor pressure (VPR) were changed by ±5%, ±10%, and ±25% for each site and each grid at the global scale, respectively. The soil carbon (SC), soil nitrogen (SN), dry deposit nitrogen, wet deposit nitrogen are changed by ±5%, ±10% and ±25%. When a variable changed at 6 levels, respectively, the rest of them were kept as the original value used for site and regional simulations. The sensitivity of model was tested by comparing the annual emissions in sensitivity simulations with the original one (Table 3).

### 2.6 Statistical Analysis

To compare the observational and simulated data, a linear regression was conducted and the slope and coefficient of determination ($R^2$) were computed. A slope less than 1 indicates the model overestimated the observation, while a slope greater than 1 means the model underestimated the observation. $R^2$ indicates how well the model captures the variation in observations. The greater $R^2$ indicates the better model performance. In addition, root-mean-square error (RMSE) was calculated to assess the difference between model simulations and observations.

### 3. Results

### 3.1 Site-level Calibration and Validation

Model slightly overestimates the observations. For all observational sites (N=81), the average $N_2O$ flux is 0.7 kg N ha$^{-1}$ yr$^{-1}$(1 kg N ha$^{-1}$ yr$^{-1}$ =0.1 g m$^{-2}$ yr$^{-1}$=0.00027 g m$^{-2}$ day$^{-1}$), with a minimum flux of 0.01 kg N ha$^{-1}$ yr$^{-1}$ (except for 0) in the dry season of African savanna, and a maximum of 5.7 kg N ha$^{-1}$ yr$^{-1}$ in tropical peatlands. Observed emissions from natural ecosystems have high variations within the same biomes, or even within several days, because environmental conditions (e.g., sudden rainfall) have significant effects on N dynamics. A linear regression between simulations and observations presents a slope of 0.72 and $R^2$ of 0.61 for all 81 sites. By removing all "0" values from tropical rainforest and temperate forests in observations, the slope decreases slightly

by 0.01 with a better $R^2$ of 0.63. The discrepancies between observation and simulation slightly decline with the RMSE changing from 0.71 to 0.608 kg N ha$^{-1}$ yr$^{-1}$ (Figure 4). A number of reasons for these differences include the sudden change of weather conditions during observation, the high uncertainty of measurement, and effect of denitrifiers especially in soils with low oxygen content. In addition, because the climate data is on a monthly step, the model did not capture the sudden changes of $N_2O$ emissions induced by extreme weather conditions at daily or sub-daily time step.

In our previous $N_2O$ emission model (Yu, 2016), the effects of climate and soil conditions were considered, but the activity of nitrifiers and its effects were not explicitly modeled. The previous model had a comparatively smaller $R^2$ and slope in comparison with observations, but overestimated $N_2O$ emissions because the model ignored the N taken up by soil microbes.

Considering major biomes, the model performs best in temperate forests ($R^2$=0.89, slope=0.64), followed by grassland and savanna ($R^2$=0.64, slope=1.05), tropical forests ($R^2$=0.52, slope=0.61) and others ($R^2$= 0.57, slope=0.51). Based on long time experimental data (longer than 6 months), the microbial trait-based model shows a better performance especially in rainforest compared to an earlier process-based model (Figure 5). The improvement on seasonal variation simulation can be partly explained by the highly active microbes in tropical areas. Compared with the tropical area with abundant precipitation, microbes contribute less to nitrogen dynamics, so the discrepancies are less significant. In other typical biomes, the trait-based model also better simulates the seasonal variations of $N_2O$ emissions. We recognized the site data in Indonesia from a cropland ecosystem converted from peatlands, which may be with higher $N_2O$ emissions than natural ecosystems in the region. This may result in relatively high emissions from this type of land ecosystems in the region.

Overall, the trait-based model better estimated total emissions and seasonal fluxes of $N_2O$ for major natural biomes (Figure 3.4). The trait-based model works better when more information of microbial activities is available to distinguish microbial guilds intra and among different biomes.

### 3.2    Model Sensitivity and Uncertainty

### 3.2.1    Model sensitivity analysis

The sensitivity analysis of model is conducted by changing climate data, soil data and N input data on three different levels and quantifying the percentage changes on model output. In our sensitivity analysis, 8 factors were

changed with 3 levels for four separate locations, representing four typical biomes. Regional analyses for each

biome type and the global scale were also conducted.

On the global scale, the model is most sensitive to air temperature, precipitation and wet deposit nitrogen.

Compare to the original model, the trait-based model has higher sensitivity to the climatic change (Table 3a). The

change of cloudiness and water vapor pressure had an indirect influence on nitrogen cycle. In most cases, $N_2O$

emissions increase with increasing temperature at observational sites (Whitehead, 1995). In our study, the emissions

varied positively with temperature. Increasing temperature by 10% enhances $N_2O$ emissions globally, but when

elevated by 25% had a negative influence on the emission. On a global scale, the precipitation change has similar

effects to the variation of temperature. Observations also indicated that the sudden precipitation change affected soil

water conditions significantly, exerting a pronounced positive influence on $N_2O$ emissions (Li et al., 2000).

Excessive rainfall showed a negative influence because soil oxygen supply is reduced by the reduction of soil pore

space. Although anaerobic soil environment favors denitrification, it reduces the respiration of oxidizing organisms

significantly, which affects the fixation and mineralization before nitrification and denitrification. The sensitivity to

SC and SN is highly related to the available nutrient to microbial activities. Abundant carbon and nitrogen energizes

nitrifiers and denitrifiers, stimulating nitrogen cycling in the soil. In general, $N_2O$ emissions positively respond to

the increase of SN and SC levels. The model is less sensitive to soil nutrient contents than to climate changes.

Overall, our analysis suggested that the trait-based model's sensitivity is similar to the earlier versions of TEM

(Zhuang et al., 2012; Qin et al., 2014) in simulating $N_2O$ emissions. The model is highly sensitive to wet N

deposition, because N deposition is an important source of soil inorganic nitrogen. In natural environment, N deposit

with rainfall (wet deposit) is about 10 times as much as that directly from the atmosphere (dry deposition)

(http://www.epa.gov/castnet/) .

At the global scale, the model is most sensitive to climatic changes. Different vegetation types have

different sensitivities and vary greatly among climatic variables (Figure 3). For all biomes, large changes with either

increasing or decreasing 20% in air temperature and precipitation have a negative effect on $N_2O$ emissions. Slight

changes by 5%, $N_2O$ emissions (increase by 8.6%) in coniferous forests are positively related to air temperature.

Tundra is most sensitive to changing air temperature with a decrease of 6.2% $N_2O$ emissions due to a 5% air

temperature decrease. Biomes in tropical and dry areas are the least sensitive to temperature variations. Biomes with

high precipitation are less sensitive. Tundra is the least sensitive biome among them, where only 0.2% emissions are

changed from a 5% change of precipitation, whereas succulent area and savanna show comparatively high

sensitivity to precipitation.

In general, model sensitivity analysis suggests that higher temperature within a certain range (15~35 °C)

means higher nitrification rate (Zhu& Chen, 2002) and denitrification rate (Stanford et al, 1975), because the growth

rate of nitrifiers is strong temperature dependent and denitrification obeys the first-order kinetic to temperature.  The

nitrification rate is influenced by the activity of ammonia-oxidizing communities. Although each guild has its own

temperature optima, the ammonia oxidation rate reaches its peak around 25~30°C (Ergruder et al., 2009; Prosser,

2011). Biomes in temperate areas are the most sensitive to temperature change. In tropical zone, the increase of

temperature negatively affects $N_2O$ emissions.

Excessive precipitation reduces the oxygen content in the soil, directly and indirectly influences the

metabolism and growth rate of nitrifiers. Biomes with high precipitation are less influenced by its variation,

compared with dry areas. This is because microbes in extreme dry conditions are more sensitive to the soil water

content. Compared to air temperature and precipitation, cloudiness and water vapor pressure are less influential,

because they have no direct effect on N dynamics in the soil. Lower cloudiness implies more solar radiation, leading

to more energy uptake by vegetation. The change of water vapor pressure is almost irrelevant to $N_2O$ emissions even

when changed by 20%.

The climate factors affect N dynamics by changing their reaction conditions, and soil factors, including soil

content and soil nitrogen content. The level of soil organic carbon and soil nitrogen shows less impact on $N_2O$

emissions (Table 3).  On a global scale, SC and SN have positive effects on N dynamics. Abundant nutrient will

keep the activity and growth rate of microbes, and consequently ensure the process of nitrification and

denitrification. Overall, $N_2O$ emissions are positively related to SC and SN inputs. Less than 3% $N_2O$ emission

changes are due to 5% to 20% changes in SC, and less than 0.3% $N_2O$ emission changes are due to 5% to 20%

changes in SN.

### 3.2.2 Key Parameters and Model Uncertainty

The parameters related to microbial guilds or vegetation biomes are chosen to conduct uncertainty analysis

(Table 4). Generally, microbes living in tropical rainforests have the highest value of Vmax, which can be partly

explained by the biological activity rate (Biederbeck et al., 1973) due to optimum temperature and moisture in the

region. Lower values appear in cold zone and dry areas, indicating a lower level of microbial activities.  For the

parameters related to microbial reactions, such as the half-saturated constant for different elements, the values vary a little between different biomes. The highest K for ammonia and nitrate appear in tropical rainforests because they have the highest soil nitrogen content. In the Briggs-Haldane kinetics and Michaelis-Menten formulation, the uptake process needs a higher K to maintain the substrate value within a reasonable range.

The percentages change in annual total $N_2O$ emissions due to changing parameters shows that the most sensitive parameters are the half saturation constants (K) associated with ammonia and nitrate uptake by microbes (Table 5). Parameter related to the growth rate of nitrifiers ($\mu_{max}$) shows the lowest sensitivity. The difference between the lowest and highest value is about 50%.

### 3.3    Global Extrapolation

During the last decade of the 20[th] century, the annual average emissions of $N_2O$ from soils were 8.7 Tg N yr$^{-1}$, with a range from 7.1 to 10.3 Tg N yr$^{-1}$. The uncertainty range of simulated $N_2O$ emissions is induced from the range of parameters shown in Table 1. The spatial pattern of the simulated global $N_2O$ emissions exhibits a large spatial variation (Figure 6). Tropical ecosystems, especially rainforests, contribute the largest fraction of the total emissions. The hotspots of emissions occurred in western Africa, South and Southeast Asia, and central Amazon Basin, which are almost the same regions of tropical rainforest. These hotspots have the optimum temperature and precipitation conditions, with rich soil organic carbon and nitrogen, stimulating the growth and metabolism of nitrifiers to increase $N_2O$ production. Except for those regions, some subtropical and temperate regions in the North Hemisphere contribute the most of the rest, including Bangladesh, South China and Central Plain of North America. Compared to tropical forests, the climate and soil conditions have significant seasonal variations. With proper temperature and precipitation, the $N_2O$ fluxes are as large as those in rainforests. These regions are usually heavily influenced by agricultural activity, and the use of fertilizers further change the pattern of $N_2O$ emissions. Some sub-polar regions also have relatively high emissions, including southern Alaska, northeastern Canada, north Scandinavia and Central Siberia. These regions are generally covered by boreal forests, having comparatively higher temperature and precipitation. The high content of organic matters provides sufficient nutrients for microbes. The regions with little precipitation and extremely low temperature have very low $N_2O$ emissions.

## 4. Discussion

### 4.1 Comparison with other estimates

Global soil $N_2O$ emissions have a large temporal variation (Figure. 7b) and a seasonal crest in August and a trough in January. The seasonal highest emissions (0.96 Tg N month$^{-1}$) are in summer of the Northern Hemisphere, with the lowest emissions (0.56 Tg N month$^{-1}$) in winter. The Northern Hemisphere and Southern Hemisphere have contrasting seasonal variations (Figure. 7a). The Northern Hemisphere contributes almost 80% of the global emissions from June to September, while emissions from the Southern Hemisphere are mainly from December to February. The global seasonal variations are similar to that in the Northern Hemisphere, suggesting that the Northern Hemisphere dominates the global annual $N_2O$ emissions (57%). Tropical regions are the most important sources from natural ecosystems, accounting for 71% of the total emissions. Temperate and Polar Regions (22%) have more emissions than in the Southern Hemisphere (7%), which is consistent with the findings of Stehfest and Bouwman (2006). Our simulations show that the emission ratios from the Northern to Southern Hemispheres are 1.5 to 1, and tropical regions (30°S-30°N) contribute 72% of the total emissions from the Southern Hemisphere.

The variation of spatial pattern is highly related to the soil and climate characteristics, as well as the vegetation types. In natural ecosystems, tropical and subtropical regions contribute the most emissions. Considering the $N_2O$ source from different biomes, it is also highly related to climate conditions and soil nutrients. Tropical forests and temperate forests are the most important sources of $N_2O$, accounting for 42% and 28% of the global total emissions, respectively. The grasslands and savannas contribute to 17% and 13% from other biomes, respectively.

Our estimated annual global $N_2O$ emissions were consistent with previous estimates. Based on three process-based models, the $N_2O$ emissions from global terrestrial ecosystems were around 8.5-9.5 Tg N yr$^{-1}$ for 1990-2000 (Tian et al., 2018). Tian et al (2015) utilized the Dynamic Land Ecosystem Model (DLEM) and estimated the $N_2O$ emissions from global land ecosystems are $12.52 \pm 0.74$ Tg N yr$^{-1}$ for 1981-2010. Huang and Gerber (2015) presented the modelled global soil $N_2O$ emission as 5.61-7.47 Tg N yr$^{-1}$ for 1970-2005. Saikawa et al. (2014) used different datasets and estimated average soil $N_2O$ emissions from 7.42 to 10.6 Tg N yr$^{-1}$ with a prognostic carbon and nitrogen (CLM-CN) - $N_2O$ model. Prentice et al. (2012) estimated that, global emissions during the 20[th] century were 8.3 - 10.3 TgN yr$^{-1}$ using DyN-IPJ dynamic global vegetation model. Using an artificial neural network approach, Zhuang et al. (2012) estimated the global $N_2O$ emissions from natural ecosystem soils were 3.37 Tg N yr$^{-1}$ for 2000. Xu et al. (2008) estimated the emissions for 1980-2000, using the relationship between $N_2O$ and $CO_2$,

were 13.31 Tg N $yr^{-1}$ with a range of 8.19-18.43 Tg N $yr^{-1}$. According to IPCC fifth Assessment Report (AR5), global $N_2O$ emissions from soils under natural vegetation varied from 3.3 to 9.0 Tg N with an average of 6.6 Tg N (Ciais et al., 2013).IPCC reported that the total emissions from anthropogenic and natural sources were 17.7 Tg N $yr^{-1}$for 1994 (Mosier et al., 1998; Kroeze et al., 1999), 9.6 Tg N $yr^{-1}$ from natural ecosystems with a range of 4.6 -

15.9 Tg N $yr^{-1}$, and 8.1 Tg N $yr^{-1}$ is anthropogenic sources with a range of 2.1 -20.7 Tg N $yr^{-1}$ (Mosier et al., 1998; Kroeze et al., 1999). Olivier et al. (1998) estimated the emission to be 10.8 Tg N $yr^{-1}$ by inverse modeling, with a range of 6.4-16.8 Tg N $yr^{-1}$. The natural emissions from IPCC Second Assessment Report (SAR) are 9 Tg N $yr^{-1}$. With a process-based model revised from DNDC (Li et al., 1992), Liu et al. (1996) estimated the global $N_2O$ emissions as 11.33 Tg N $yr^{-1}$. Carnegie- Ames- Stanford Approach gave a global estimation of 6.1 Tg N from soil

surface (Potter et al., 1996). Prinn et al. (1990) estimated the total emission for 1978-1988 as 20.5±2.4 Tg N $yr^{-1}$ using a 9-box model. Their estimates included natural and anthropogenic sources, so the total value was significantly larger. The slightly lower estimate of $N_2O$ in our study may be due to the consideration of microbial consumption of nitrogen, and the ignorance of N fixation from symbiotic system (Rochette et al., 2005; Zhong et al., 2009; Shah, 2014).

**4.2      Major Controls to soil $N_2O$ Emissions**

In our simulation, the emission was primarily controlled by soil temperature, soil moisture, soil nutrient content, and nitrogen deposition. The highest $N_2O$ emissions are usually due to high temperature and ample precipitation, because increasing soil temperature stimulates microbial activities related to nitrification and denitrification.

Increased temperature within a threshold was generally assumed to enhance the microbial activity (Biederbeck and Campbell, 1973), to increase the nitrification and denitrification rate, and generally to increase the $N_2O$ fluxes on annual scales. The response of microbial activity is greatly affected by temperature but the situation is complex because both the growth rate and respiration component is large. Generally, the respiration rate increases over temperature and the optimum temperature for bacterial growth is around 25-35°C (Pietikäinen, 2005), although

for some nitrifiers the optimum temperature lies at 42°C (Painter, 1970). Studies on the nitrification rate have shown a similar trend by temperature. The optimum temperature ranges between 20°C and 35°C. Below 20°C, the nitrification-denitrification rate drop sharply and there is almost a linear relationship between them. The situation is similar when temperature is above 35°C and the decreasing rate is larger than the increasing rate below 20°C. This is

consistent with our sensitivity analysis for different biomes, which indicates that vegetation types in temperate regions were more sensitive to temperature changes than tropical regions. The original temperature in temperate region is likely to be lower than the optimum temperature range, a slightly increase in temperature will thus increase $N_2O$ emissions. Lab experiments show that the increase of temperature has positive impacts on $N_2O$ emissions, although less significant than the prediction using the Arrhenius equation (BassiriRad, 2000; Zhu and Chen, 2002; Schindlbacher et al, 2004).

Precipitation is significantly correlated with soil moisture, which strongly influences the microbial activity (Zhao et al., 2016; Castro et al., 2010), affects the soil oxygen diffusion (Neira et al., 2015). Rainfall also determines the amount of wet N deposition (Vet et al., 2014), and consequently influences the $N_2O$ emissions. In our sensitivity analysis, increased precipitation and wet deposition were simulated to initially promote the nitrification and denitrification rate, and $N_2O$ emissions. Decreasing precipitation and wet deposition have a negative effect on a global scale. However, excessive precipitation inhibits nitrification, because oxygen acts as the electron acceptor in this process. Lower water content may limit the nitrifying bacterial activity by restricting substrate supplies and reducing hydration and activity of enzymes (Stark and Firestone, 1995). When the soil becomes partially anaerobic with very high water content, nitrifiers will be highly inhibited and most emissions are due to denitrification process. The influence of precipitation is similar to the effects of temperature (Klemedtsson et al., 1988). The highest $N_2O$ production appears within an optimum range of soil moisture levels. The rate increases below the optimum range and sharply decreases with extremely high precipitation. These findings are consistent with previous results (Li et al, 1992; Liu et al, 1996; Prentice et al, 2012; Saikawa et al, 2013). Biome with dramatic seasonal precipitation changes shows high sensitivity to the change of precipitation, including savanna and temperate grassland. This is consistent with the experimental study, suggesting that rewetting after extreme drought causes a rapid increase of $N_2O$ emissions especially in the initial rewetting stage (Guo et al., 2014).

In our simulation, the change of soil nutrient content did not lead to a significant change of $N_2O$ emissions. Increasing or decreasing soil carbon content by 10% resulted in 1.5%~1.6% change in emissions (Table 3a), which is not as sensitive as the climate conditions. The effect of soil nutrient is complex. Elevated soil carbon availability influences microbial activities. Soil microbial nitrogen uptake and growth rate is regulated by soil carbon content, especially in a carbon limited state (Farrell et al., 2014). Carbon acts as substrate in denitrification and elevated carbon is expected to enhance the $N_2O$ emissions (Holmes et al., 2006). In the meantime, elevated soil carbon

content will increase plant carbon productivity, which further increases the consumption of soil nitrogen. Plants and microorganisms compete for nitrogen in many processes. The increase of plant production may decrease the availability of nitrogen, and consequently inhibit the $N_2O$ emissions (Zhu et al., 2017).

**4.3      Model limitation and Implication for future studies**

There are a number of limitations of this study.  First, our simulation uncertainty is from model parameterization and uncertain structure due to the incomplete understanding of the processes (Janssen et al., 1994). Current parameter values for microbial guilds area mainly come from semi-empirical experiment results, including the measurements in experiments or observations.  But these are limited by available observational data: one set of parameter was applied for all biome grids and ignored the microbial diversity in grids with the same biome. Our

current trait-based model did not consider nitrogen input from symbiotic and non-symbiotic N fixation, because some $N_2O$ emissions may be attributed to N fixation (Cosentino et al., 2015; Flynn et al., 2014; Shah, 2014; Zhong et al., 2009). At the global scale, N input through nitrogen fixation is comparable to the input through N deposition. However, there is a large variation existing between land use types, led by the distribution of related bacterial and plants. The contribution of N fixation to total $N_2O$ emission is not considered in this study. In addition, the model

has not considered the microbial effect on denitrification, which is also an essential process not only under aerobic but also under anaerobic conditions. The effect of denitrifying bacteria is a more complicated problem compared to nitrification. By introducing the effect of denitrifying bacteria will establish a more completed relation between carbon and nitrogen.

          Second, uncertain forcing data including climate, soil conditions, and microbial guild assumptions and

observational data could also bias our estimates. Significant uncertainty remains for input data, especially for several eco-physiological factors of soil microbes. Climate data and soil data were collected from different sources at 0.5° x 0.5° resolution, which may not be suitable for a certain site.

          Third, some regions (e.g., North America and Europe) have rich observational data to parameterize the model. Compared to tropical rainforests and temperate forests, observational data from tundra and wet tundra are far

less. Further effort on improving observational accuracy and enriching data especially in polar zones would improve the performance of future models.

**5.	Conclusions**

Most existing process-based models of soil $N_2O$ emissions have not considered the effect of the detailed microbial dynamics in a spatially and temporally explicit manner. This study developed and applied a trait-based biogeochemistry model to estimate the global seasonal and spatial variations through the last decade of the 20[th] century. The major source of $N_2O$ was found to be tropical and temperate forests. The spatial and temporal variation was largely caused by the distribution of microbial traits, soil carbon and nitrogen sizes, as well as different precipitation and temperature regimes. The global soil $N_2O$ emissions from global natural ecosystems were estimated to be 8.7 Tg N yr$^{-1}$ on average. Our study suggested that more experimental data on microbial ecophysiology and $N_2O$ fluxes shall be collected to improve future quantification of $N_2O$ emissions from global natural ecosystem soils.

**6.	Acknowledgment**

This study is supported through projects funded by the NASA Land Use and Land Cover Change program (NASA-NNX09AI26G), Department of Energy (DE-FG02-08ER64599), the NSF Division of Information & Intelligent Systems (NSF-1028291). Thanks to Rosen Center for Advanced Computing (RCAC) at Purdue University for computing support.

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

Table 1: Variables and Model Parameters used for microbial traits

| Parameters | Description | Units | Values |
|---|---|---|---|
| $V_{ox}^{NH3}$ | Daily ammonia losses from oxidation | g N m$^{-2}$ day$^{-1}$ | |
| $V_{ox}^{NO2}$ | Daily nitrite losses from oxidation | g N m$^{-2}$ day$^{-1}$ | |
| $V_{AOO}^{NH3}$ | Daily ammonia uptake into biomass of ammonia-oxidizing organism (AOO) | g N m$^{-2}$ day$^{-1}$ | |
| $V_{NOB}^{NH3}$ | Daily ammonia uptake into biomass of nitrite-oxidizing bacteria (NOB) | g N m$^{-2}$ day$^{-1}$ | |
| $D_O^{NO}$ | Daily biomass loss due to the detoxification of NO by the ammonia-oxidizing bacteria (AOB) and NOB mediated reactions | g N m$^{-2}$ day$^{-1}$ | |
| $D_O^{NO2}$ | Daily biomass loss due to the detoxification of NO$_2$ by the AOB and NOB mediated reactions | g N m$^{-2}$ day$^{-1}$ | |
| $V_{max}^{NH3}$ | The maximum ammonia uptake rate | mol L$^{-1}$ day$^{-1}$ | 0.24-1.04 |
| $K_M^{NH3}$ | Ammonia inhibition constant for AOO | $\mu$ mol L$^{-1}$ | 1.9-61 |
| $K_M^{NO2}$ | Nitrate inhibition constant for NOB | $\mu$ mol L$^{-2}$ | 25-260 |
| $K_M^{O2}$ | Oxygen inhibition constant for AOO | $\mu$ mol L$^{-2}$ | 1.4-23 |
| $B_{TA}$ | Total biomass of AOO, including biomass carbon ($B_C$) and biomass nitrogen ($B_N$) | g N m$^{-2}$ | |
| $B_{TN}$ | Total biomass of NOB, including biomass carbon ($B_C$) and biomass nitrogen ($B_N$) | g N m$^{-3}$ | |
| $\mu_{max}$ | The maximum growth rate for nitrifiers | day$^{-1}$ | 0.01-0.09 |
| $d_B$ | Cell division of NOB and AOO | | |
| Q | Cellular Quota for nitrogen ($Q_N$) and carbon ($Q_C$) | | |

Table 2: Site information of observational data used for model calibration and validation

| Site name | Ecosystem Type | longitude | latitude | Temperature (°C) | Precipitation (mm) | length of experiment | $N_2O$ Fluxes (kgN ha$^{-1}$ yr$^{-1}$) | Reference |
|---|---|---|---|---|---|---|---|---|
| Kauri Creek, Australia | Rainforest | 145.5 | -17.5 | 17.6-23.9 | 25.5-252.3 | 10~19 | 0.03-0.035 | Breuer et al.(2000) |
| Lake Eacham, Australia | Rainforest | 145.5 | -17 | 20.2-27.1 | 42.2-309.3 | 8~22 | 0.02~0.09 | Breuer et al.(2000) |
| Massey Creek, Australia | Rainforest | 145.5 | -17.5 | 19.0-24.3 | 69.7-236.1 | 10~18 | 0.07~0.20 | Breuer et al.(2000) |
| Chagurarama, Guarico State, Venezuela | Savanna (grassland) | -79.5 | 36.5 | 3.5 | 104.8(dry season) | 9 | 0.01 | Hao et al.(1988) |
| 10km from No 4 | Savanna (woodland) | -79.5 | 36.5 | 3.5 | 104.8 | 9 | 0.03 | Hao et al.(1988) |
| Lake Creek, Linn County Williamette Valley, Oregon | Grass | -123.5 | 44.5 | 10.7 | 305.7 | 93 | 0.31 | Horwath et al (1998) |
| Höglwald, Germany | Coniferous | 14 | 51 | 14.6 | 66.8 | 30 | 0.04~0.12 | Butterbach-Bahl et al(1997) |
| Kiel, Germany | Deciduous | 112.5 | 23 | 21.4 | 1927 | 365 | 0.4~4.9 | Mogge et al.(1998) |
| Mainz, Germany | Grass | 8.5 | 50 | 10 | 45.5-546 | 32-71 | 0.02-0.13 | Seiler and Conrad(1981) |
| Ballyhooly, Republic of Ireland | Coniferous | -8.5 | 52 | 9.6 | 89.9 | 3 | 0 | Butterbach-Bahl et al(1998) |
| Poppel, Belgium | Deciduous | 5 | 51.5 | 11 | 657-1017.6 | 317-365 | 0 | Goossens et al.(2001) |
| Central Scotland | Deciduous | -4.5 | 56.5 | 8.7 | 828.8 | 210 | 1.15~2.29 | Pitcaim et al. (2002) |
| Guanica Commonwealth Forest, SW Puerto Rico | Tropical Dry forest | -63 | -10 | 25.6-26.3 | 108.4-1626.4 | 153-365 | 0.02-0.7 | Erickson et al (2002) |
| San Dimas Experiment Forest | Mediterranean Shrub lands | -118 | 34 | 13-42 | 696 | 60 | 0.05~0.15 | Anderson and Poth (1989) |
| Lincolen Canterbury, New Zealand | Grassland | 172.5 | -43.5 | 1~21 | 2-47 | 400 | 0.255 | Müller and Shelock (2004) |
| Nylsvley Nature Reserve, South Africa | Savanna | -24.5 | 28.5 | 12~15 | 625 | 19 | 0.28 | Scholes et al.(1997) |
| Gambutt, South Kalimantan, Indonesia | Tropical Peatland | 114.5 | 3.5 | 28 |  | 28 | 5.698 | Hadi et al. (2000) |
| Barambai, South Kalimantan, Indonesia | Tropical forest | 114.5 | 3 | 28 |  | 28 | 2.628 | Hadi et al. (2000) |
| Fazenda Victória, Brazil | Tropical rain forest | 55 | 3 |  | 1800 | 1.5 yr | 2.6 | Davidson et al (2004) |
| Orinoco Ilanos, Venezuela | Savanna | -63.5 | 9.5 | 27.3 | 992 | 1 yr | 0.73 | Simona et al (2004) |

| Location | Ecosystem type | | | | | | | Reference |
|---|---|---|---|---|---|---|---|---|
| Horquetas, Costa Rica | Tropical Pastures | -85 | 10 | 25.8 | 3962 | 23-30 | 2.365 | Veldkamp et al. (1998) |
| La Selva biological station, Costa Rica | Tropical Forest | -84 | 10.5 | 25.8 | 3962 | 15months | 3.74 | Keller& Reiners (1994) |
| Isabela and Mayaguez, Puerto Rico | Tropical Grassland | -67 | 18 | | | 1yr | 1.51 | Mosier& Delgado (1997), Mosier(1997a) |
| Luquillo Experimental Forest, Puerto Rico | Subtropical wet Forest | -66 | 18 | 23.5-27 | 2900-3200 | 1yr | 1.75 | Erickson et al (2001) |
| Kilauea, Hawaii | Rain forest | -155.5 | 19.5 | | | 1yr | 0.223 | Riley & Vitousek (1995) |
| Wudaoliang, Qinghai, China | Alpine Grassland | 93 | 35 | -5.6 | 200-400 | 1yr | 0.069 | Pei (2003) |
| Mount Taylor, New Mexico | Temperate forest | -107.5 | 35.5 | (-10.5) to 17 | 640-720 | 2yrs | 0.03  0.09 | Matson et al (1992) |
| Nevada Desert FACE facility, US | Desert | -116 | 37 | | 140 | 2yrs | 0.11 | Billings et al.(2002) |
| Colorado, USA | Temperate Grassland | -104.5 | 40.5 | (-5) to 30 | 350 | 3yrs | 0.167 | Mosier& Delgado (1997), Mosier(1997a) |
| Changbai Mountain Forest Research Station | | 127 | 41.5 | (-7.3) to 3.3 | 700-1300 | 138 | 0.28 | Chen et al (2000) |
| Browns Park Formation, Wyoming, USA | SageBrush Steppe | -107 | 41.5 | 2.7 | 525 | 2 yrs | 0.21 | Matson et al (1991) |
| Harvard Forest,  USA | Temperate Forest | -72 | 42.5 | (-8) to 23 | | 2yrs | 0.02  0.06 | Bowden et al(1990) |
| Whiteface Mt, NY, USA | Temperate Forest | -74 | 44.5 | 0~18 | | 1990 | 0.185 | Castro et al (1992) |
| Mt Mansfield, VT, USA | Temperate Forest | -73 | 44.5 | 4~19 | | 1990 | 0.1708 | Castro et al (1992) |
| Mt Ascutney, VT, USA | Temperate Forest | -72.5 | 43 | 6~22 | | 1990 | -0.098 | Castro et al (1992) |
| Mt Washington, NH, USA | Temperate Forest | -71 | 44 | 4~17 | | 1990 | -0.02 | Castro et al (1992) |
| Acadia, ME, USA | Temperate Forest | -68.5 | 44 | 5~20 | | 1990 | 0.0315 | Castro et al (1992) |
| Waldhausen, Germany | Temperate Forest | 10 | 49 | (-0.4)~20 | | 1981-1982 | 0.473 | Shmidt et al. (1988) |
| Bechenheim, Germany | Temperate Forest | 8 | 49.5 | 0~18.3 | | 1981-1982 | 0.802 | Shmidt et al. (1988) |
| Langenlonsheim, Germany | Temperate Forest | 8 | 50 | 0~18.6 | | 1981-1982 | 0.714 | Shmidt et al. (1988) |

Table 3: Sensitivity Studies of $N_2O$ emissions (%) responding to changes of: (a) climate and soil data at different levels; (b) temperature at 5% and 20% for different vegetation types; (c) precipitation at 5% and 20% for different vegetation types

(a)

|  | 5% | -5% | 10% | -10% | 20% | -20% |
|---|---|---|---|---|---|---|
| Air temperature | 3.2 | -2.5 | 1.2 | -5.5 | -11 | -17 |
| Precipitation | 4.5 | -1.8 | 0.97 | -3.4 | -6 | -10 |
| Cloudiness | -0.85 | 0.43 | -3.2 | 1.1 | -5 | 0.9 |
| Water Vapor Pressure | 0.03 | -0.015 | 0.07 | -0.032 | 0.1 | -0.92 |
| Soil Carbon | 0.8 | -0.7 | 1.5 | -1.6 | 2.9 | -3.2 |
| Soil Nitrogen | 0.2 | -0.17 | 0.24 | -0.25 | 0.27 | -0.3 |
| Dry Deposit N | 0.18 | -0.23 | 0.65 | -0.60 | 3.5 | -2.4 |
| Wet Deposit N⁻ | 7.2 | -8.5 | 18 | -17 | 33 | -29 |

(b)

|  | 5% | -5% | 20% | -20% |
|---|---|---|---|---|
| Tropical Forest | -1 | -0.5 | -19 | -11 |
| Temperate Evergreen Forest | 6.5 | -4 | -6 | -13 |
| Temperate Deciduous Forest | 4.3 | -5.5 | -7 | -15 |
| Temperate Coniferous Forest | 8.6 | -4.2 | 3 | -37 |
| Temperate Grassland | 2.1 | -3.5 | -11 | -19 |
| Savanna | 0.5 | -2 | -16 | -7.2 |
| Succulent | -2 | -0.2 | -24 | -5.5 |
| Mediterranean Shrub lands | 0.7 | -1.5 | -17 | -12 |
| Tundra | 5.5 | -6.2 | 3.5 | -27 |

(c)

| | 5% | -5% | 20% | -20% |
|---|---|---|---|---|
| Tropical Forest | 0.7 | -0.3 | -11 | -12 |
| Temperate Evergreen Forest | 2.6 | -3.5 | -8.2 | -12 |
| Temperate Deciduous Forest | 4.2 | -0.8 | -9 | -8 |
| Temperate Coniferous Forest | 1.5 | -2.2 | -5.3 | -9.7 |
| Temperate Grassland | 4.6 | -3.3 | -2.6 | -12 |
| Savanna | 5.7 | -2.8 | -5.3 | -17 |
| Succulent | 4.4 | -6.3 | -2.7 | -18 |
| Mediterranean Shrub lands | 2.2 | -3.7 | -6.5 | -15 |
| Tundra | 0.2 | -0.2 | -3.1 | -11 |

Table 4: Key parameters' values after calibration

| | Vmax_AOO (M day$^{-1}$) | Vmax_NOB (M day$^{-1}$) | miu_max (day$^{-1}$) | K_NH ($\mu$M) | K_NO ($\mu$M) | K_O ($\mu$M) |
|---|---|---|---|---|---|---|
| Tropical Forest | 0.54 | 3.5 | 0.06 | 56 | 100 | 6.8 |
| Temperate Evergreen Forest | 0.52 | 3 | 0.05 | 46 | 90 | 7.2 |
| Temperate Deciduous Forest | 0.5 | 3 | 0.05 | 48 | 88 | 7 |
| Temperate Coniferous Forest | 0.52 | 3.2 | 0.05 | 46 | 82 | 7 |
| Temperate Grassland | 0.5 | 2.5 | 0.05 | 38 | 60 | 12 |
| Savanna | 0.5 | 2.5 | 0.04 | 42 | 62 | 12 |
| Succulent | 0.46 | 1 | 0.04 | 22 | 52 | 14 |
| Medeterranean Shrub lands | 0.48 | 2 | 0.04 | 40 | 66 | 14 |
| Tundra | 0.48 | 2.5 | 0.05 | 40 | 68 | 4.2 |

Table 5: Sensitivity (%) of key parameters for biomes

| | 5% | -5% | 25% | -25% |
|---|---|---|---|---|
| Vmax_AOO (M day$^{-1}$) | 1.3 | -3.1 | 7 | -9.9 |
| Vmax_NOB (M day$^{-1}$) | 0.8 | -2 | 5.5 | -7.5 |
| miu_max (day$^{-1}$) | 2.2 | -1.3 | 8.7 | -9.7 |
| K_NH (µM) | -0.25 | 0.26 | -0.52 | 0.38 |
| K_NO (µM) | -0.15 | 0.28 | -0.17 | 0.3 |
| K_O (µM) | -0.23 | 0.24 | -0.14 | 2 |

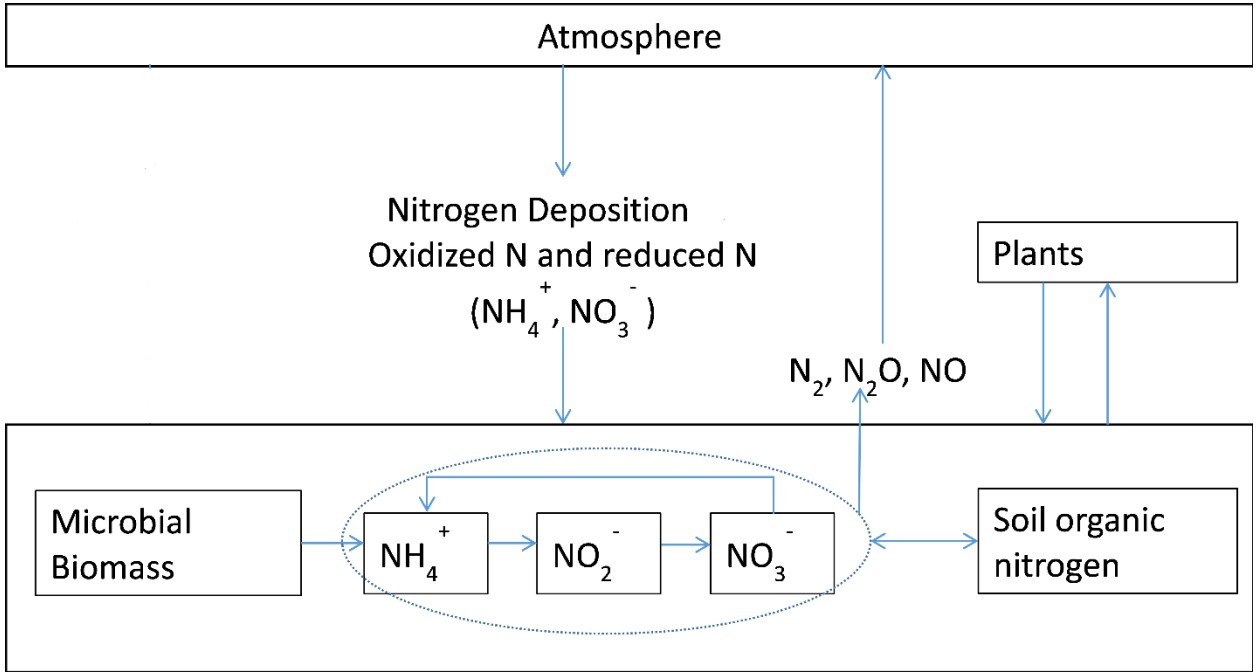

Figure 1: Schematic diagram of N₂O emissions and N cycling between plants, soils, and the atmosphere: The input

of N from the atmosphere to soils through nitrogen deposition as nitrate and ammonia; microbial biomass dynamics

were modeled; Nitrification is modeled as a function of microbial biomass, soil organic nitrogen, and physical

conditions, more details refer to Yu (2016); N uptake by plants is modeled in original TEM (McGuire et al., 1992).

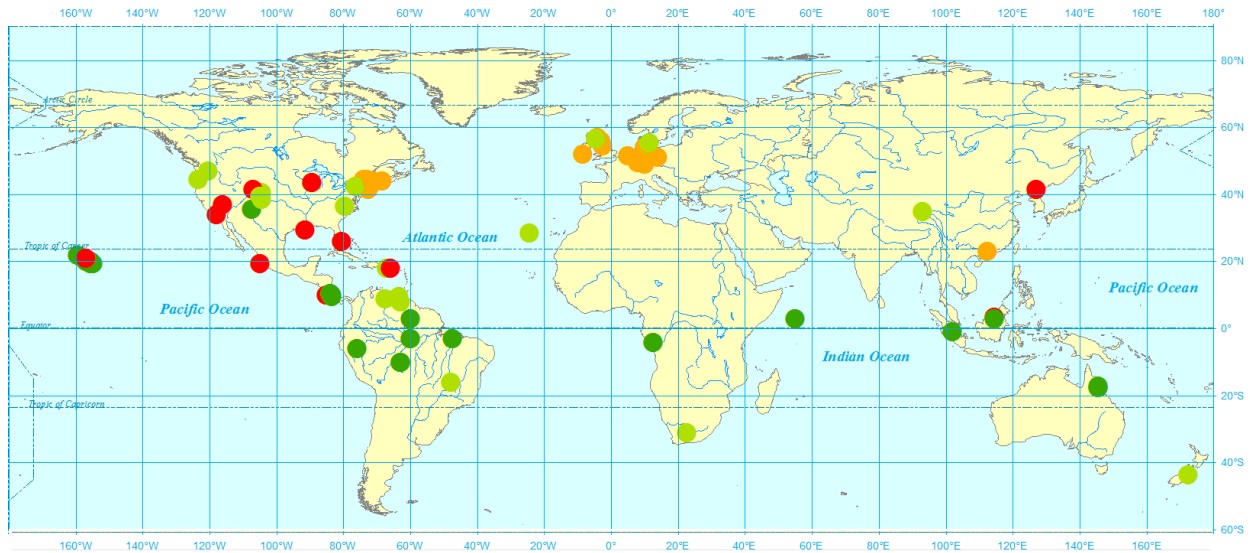

Figure 2: N$_2$O observational sites used in this study: tropical forest (dark green), grassland (light green), temperate forest (yellow), others (red).

(a)

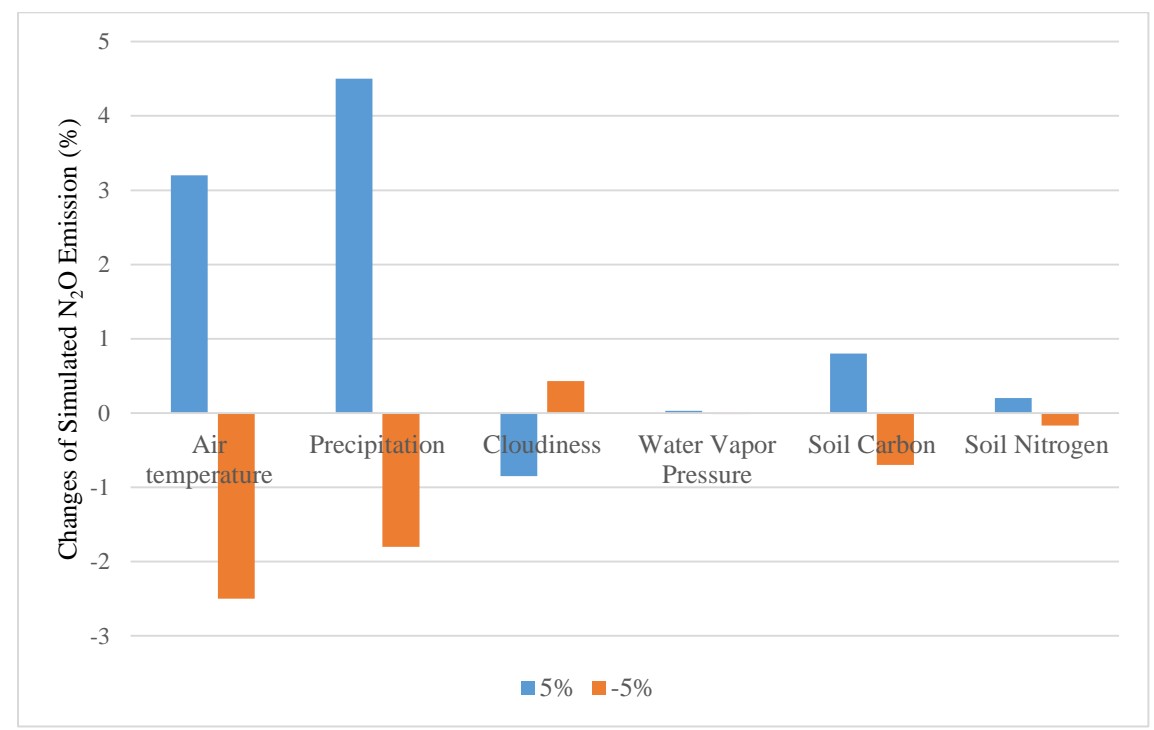

(b)

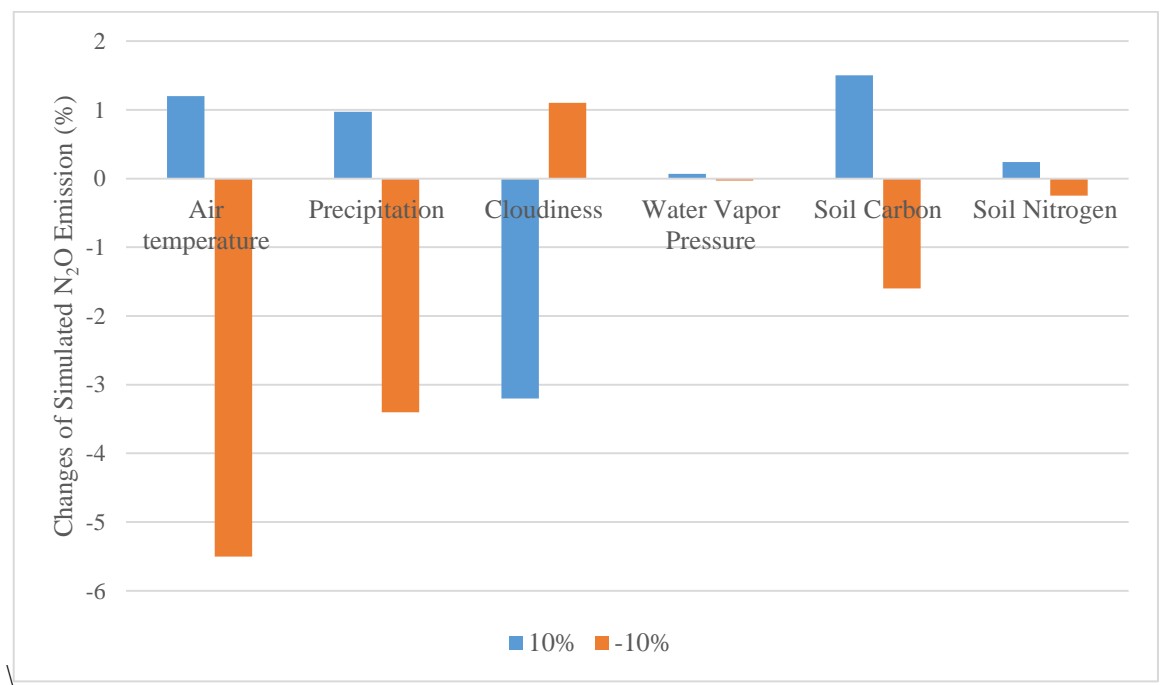

(c)

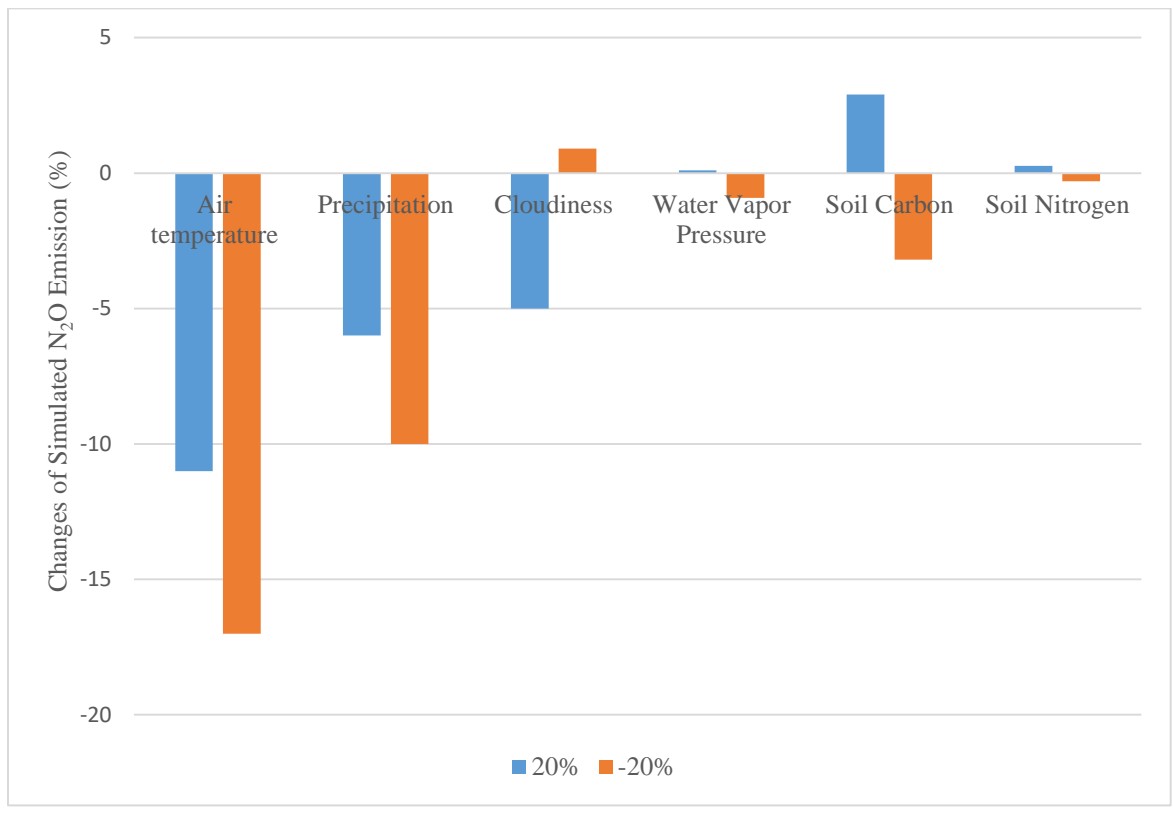

Figure 3: The sensitivity study of N$_2$O emissions in natural terrestrial ecosystems by changing different climate variables by: (a) 5%; (b) 10%; (c) 20%

(a)

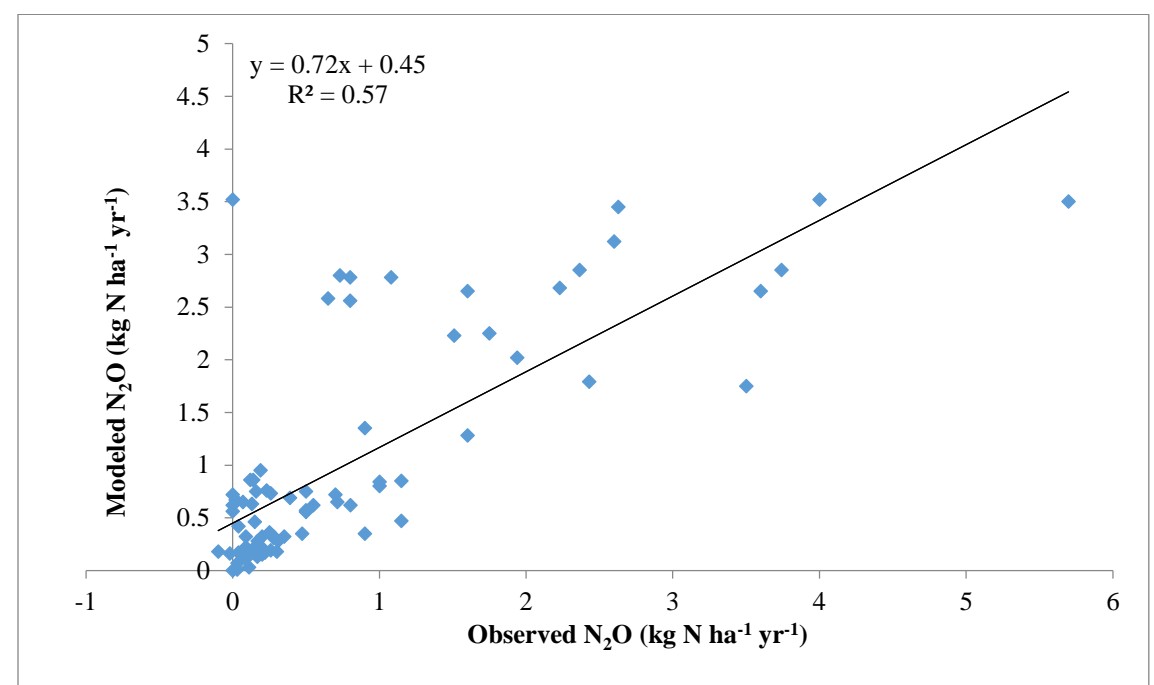

(b)

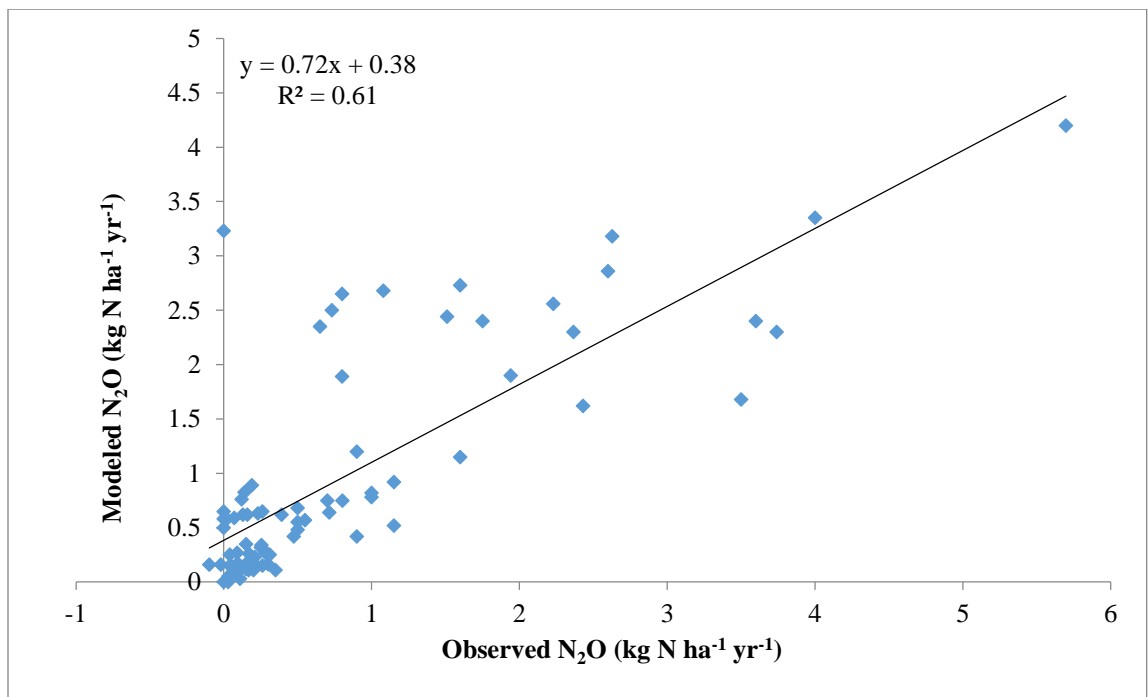

(c)

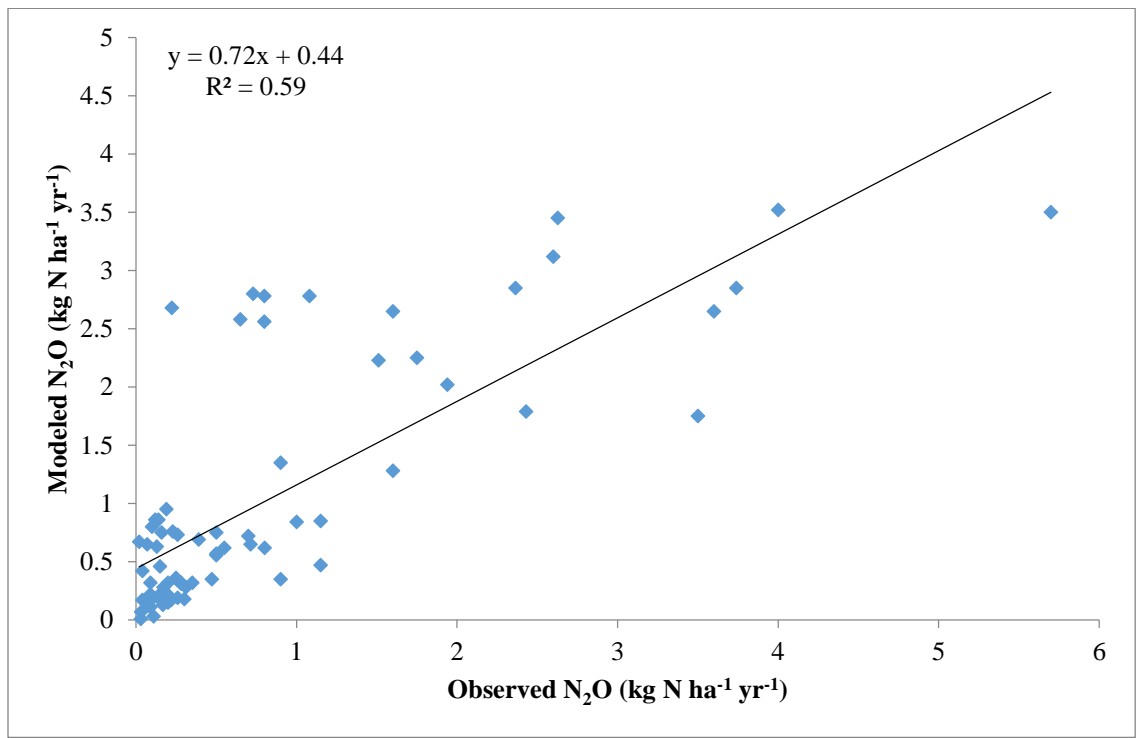

(d)

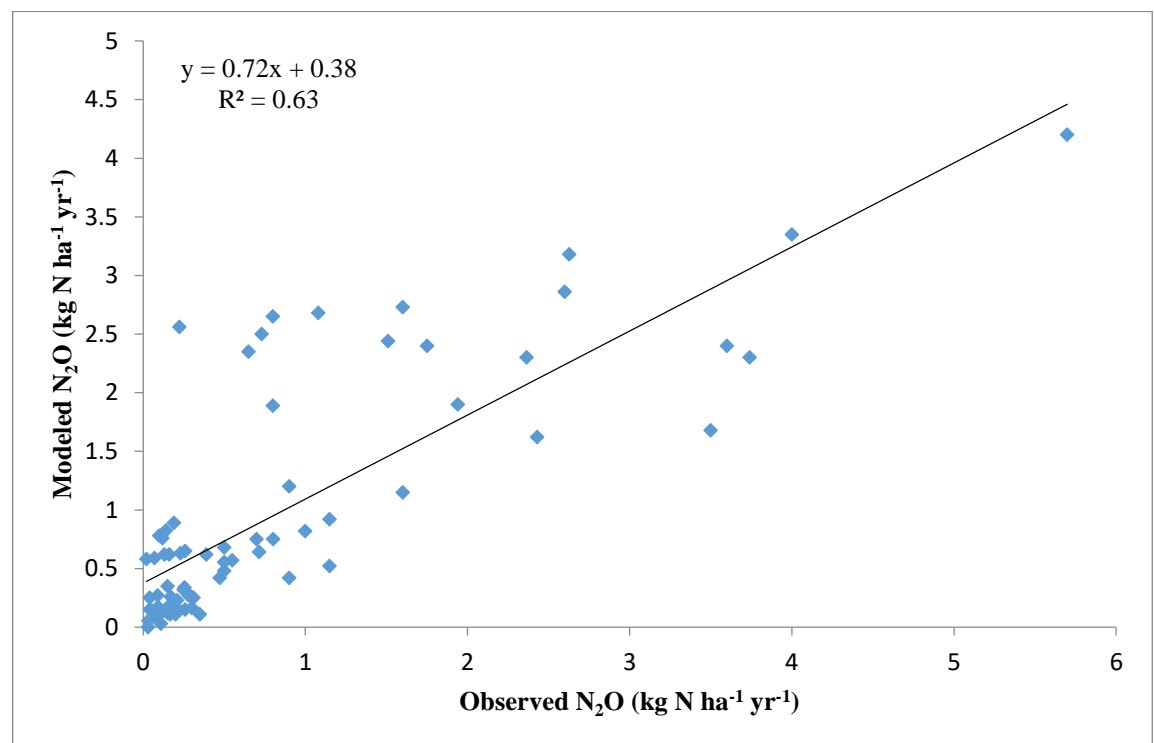

Figure 4: Observational and model simulation of annual $N_2O$ emissions (a) with all observational data and original process-based model TEM (Yu, 2016); (b) With all observational data and microbial trait-based model; (c) Without observational "0" and original process-based model; (d) Without observational "0" and microbial trait-based model.

(a)

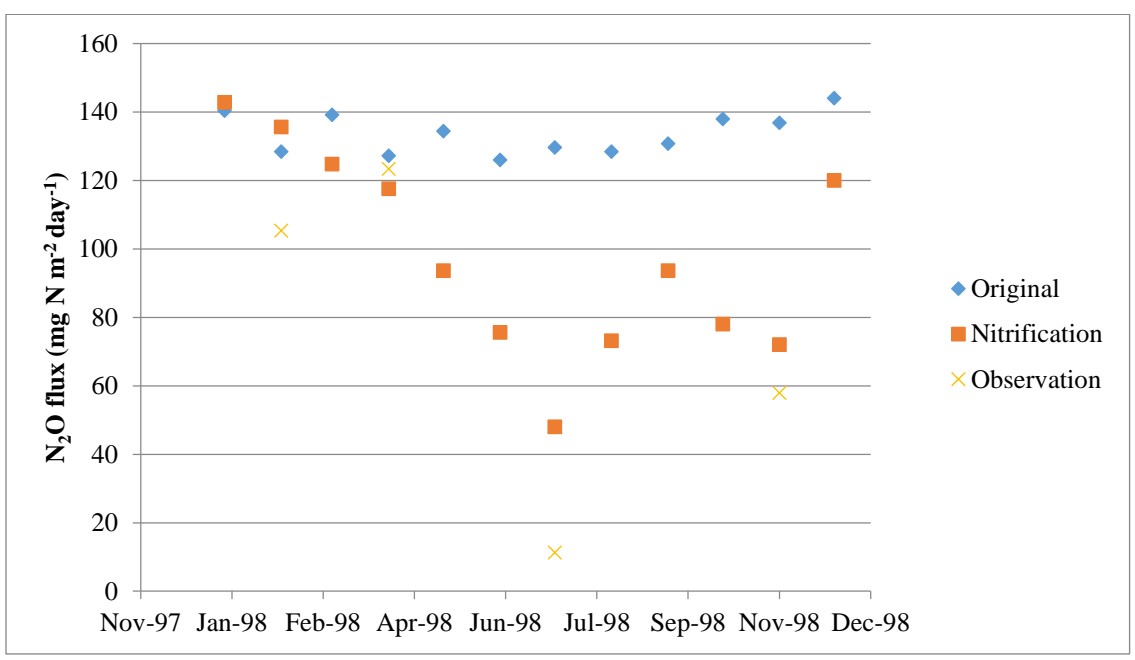

(b)

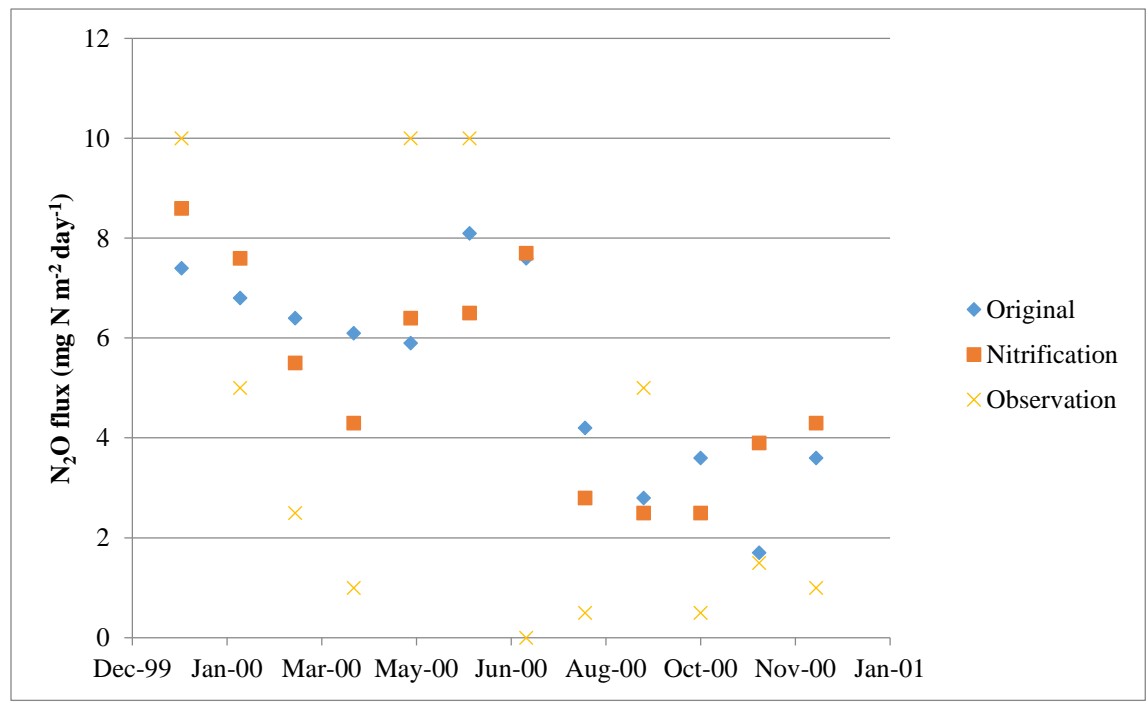

(c)

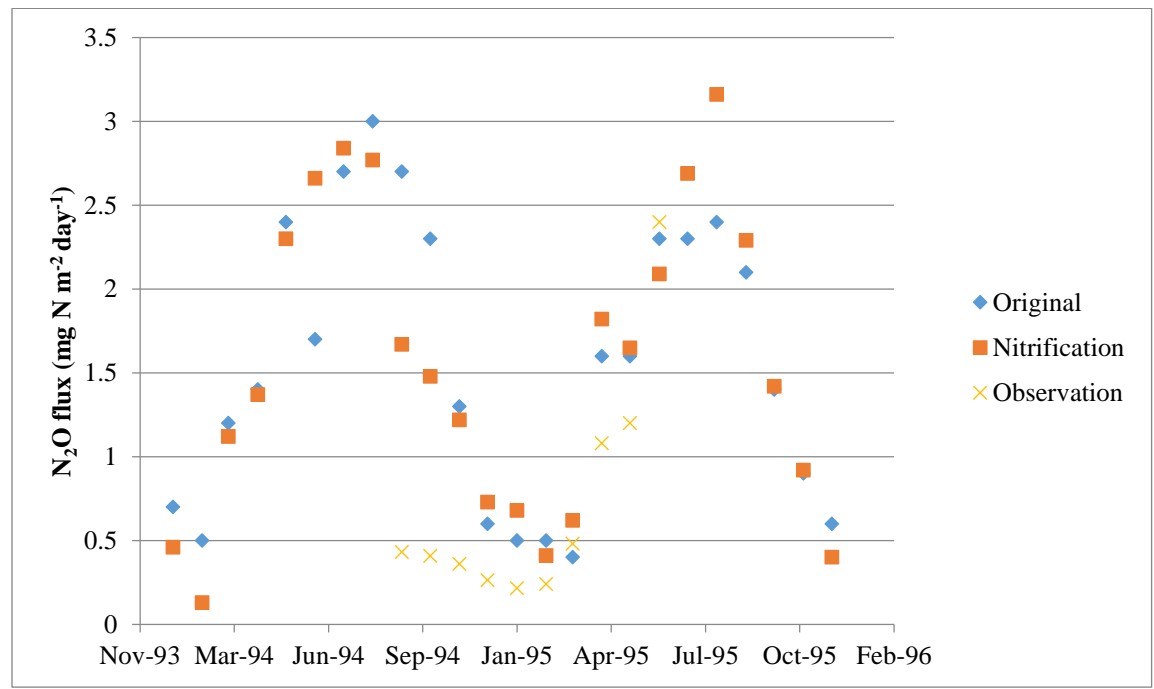

(d)

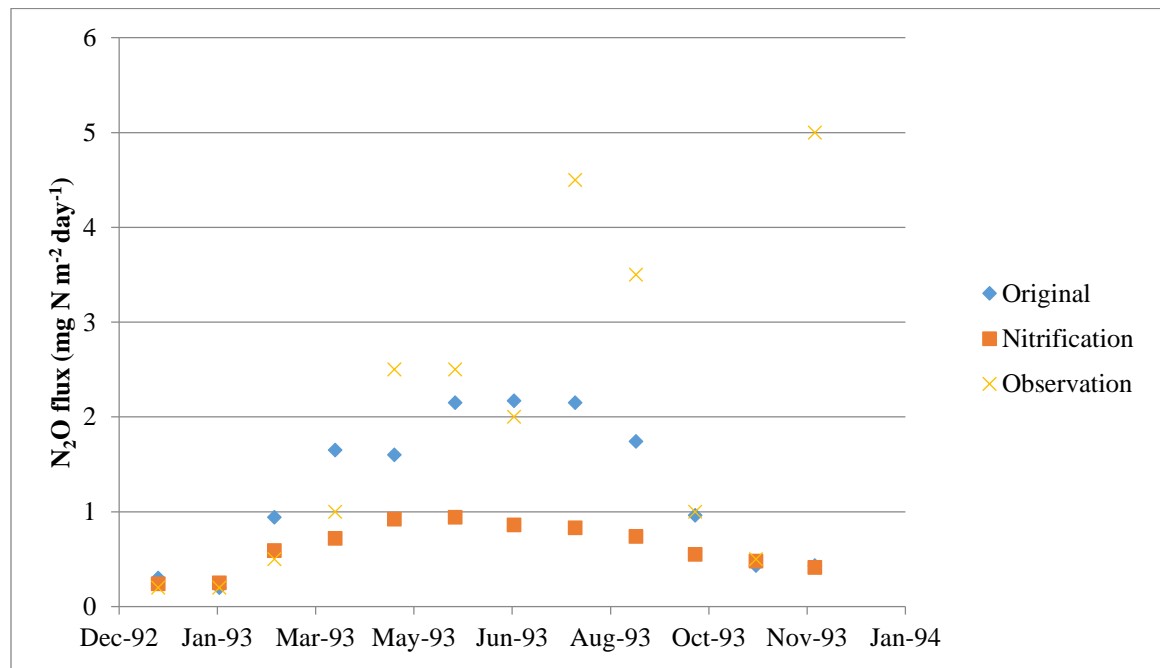

Figure 5: Model Validation at: (a) Rainforest: 145.5°E, 17.5°S; (b) Grassland: 172.5°E, 43.5°S; (c) Coniferous: 14°E, 51°N; (d) Deciduous: 10°E, 54°N

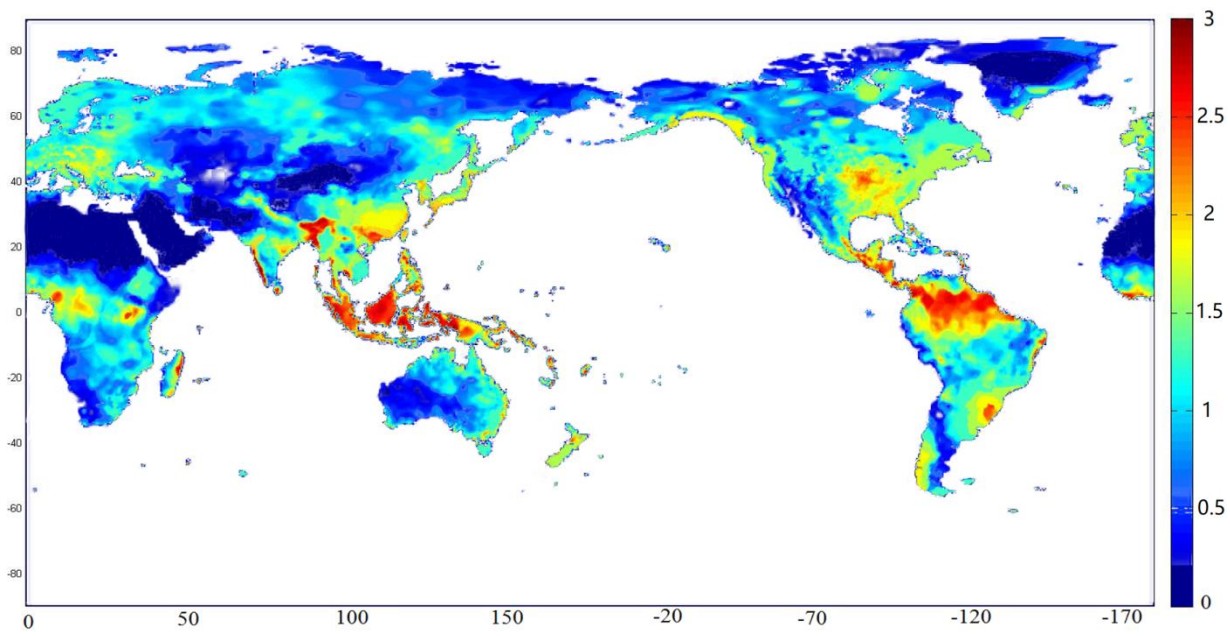

Figure 6: Spatial patterns of N$_2$O emissions (kg N ha$^{-1}$ yr$^{-1}$) from natural ecosystems (1990-2000)

(a)

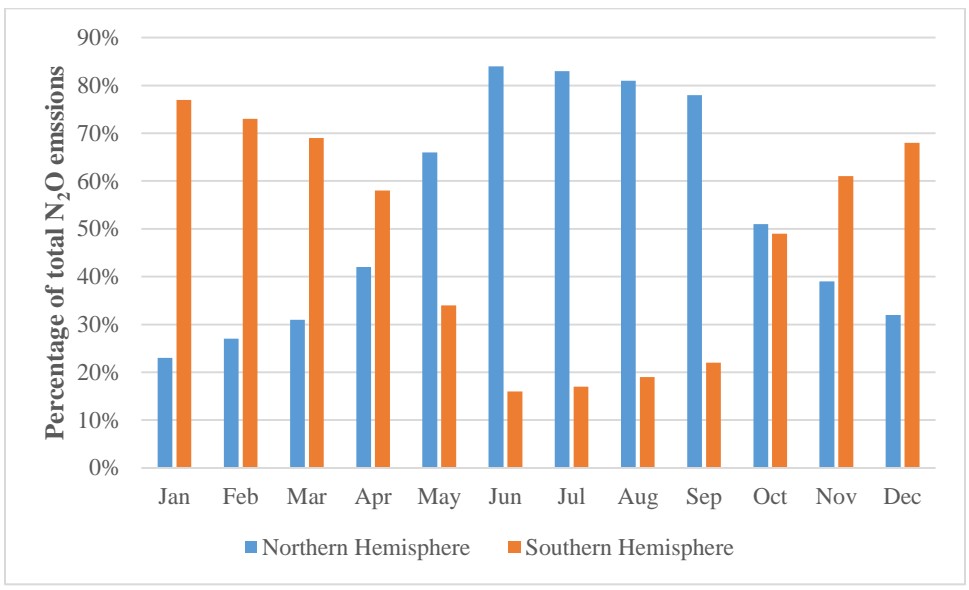

(b)

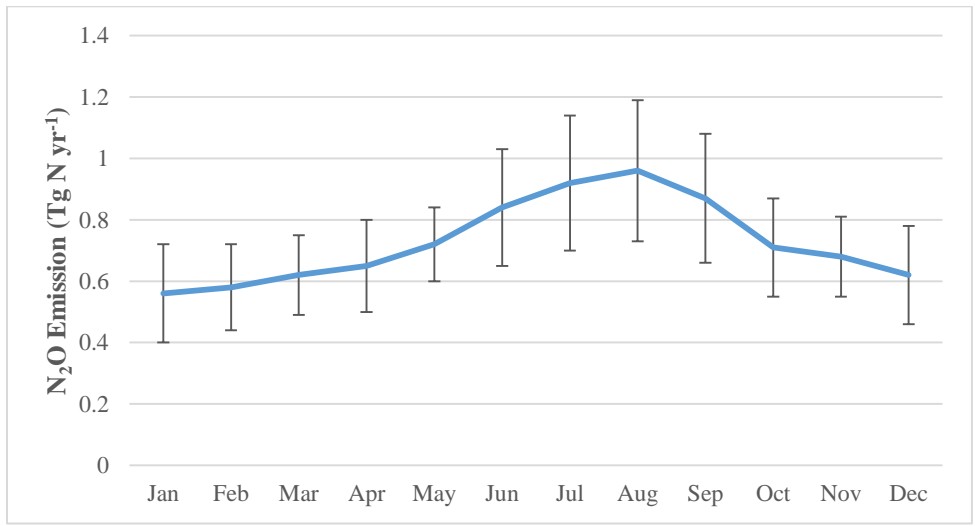

Figure 7: Seasonal variation of $N_2O$ emissions: (a) Contribution of the Northern and Southern Hemisphere; (b) Global average monthly emissions and their standard deviations for the period 1990-2000 (Tg N yr$^{-1}$).