# Peer review of "Quantifying Global N2O Emissions from Natural Ecosystem Soils Using Trait-Based Biogeochemistry Models"

_Biogeosciences, 2018_

## Referee Comment (RC1) · Anonymous Referee #1 · 30 Oct 2018

General Comments: This study tried to improve their previous first-order module within the TEM model through incorporate microbe trait to simulate nitrous oxide (N2O) fluxes from the natural soils. The results indicated that a total emission of N2O was 8.7±1.6 Tg N yr-1 globally, and 42% of this emission was attributed to tropical forest. They found that the average N2O flux is 0.7 kg N ha-1 yr-1, with a minimum flux of 0.01 kg N ha-1 yr-1 in the dry season of African savanna, and a maximum of 5.7 kg N ha-1 yr-1 in tropical peatlands based on all observational sites (N=81). Compared with their previous version, the current microbial trait-based model shows a better performance, especially in rainforest, because of the consideration of N taken up by soil microbes. They concluded that new model captured more variations of N2O emission in response

to seasonal changes in climate. However, there are substantial weaknesses in this study that should be addressed for making an incremental advancement in modeling N2O emissions from soils.

Specific Comments: 1. The authors intent to estimate N2O emission from natural soils during 1990-2000. However, the terrestrial ecosystems have been extensively disturbed and managed. It's unclear about what the natural soils mean in this manuscript. There is no detail information on how they generate the natural ecosystem data across the global land surface. 2. Global map (Fig. 5) shows N2O emissions from the cultivated areas where crops planted during 1990-2000 according to my knowledge. Do you consider background emission from cropland as natural emissions? Or you treat cropland as other types of vegetation? 3. It needs to provide more explicit explanation on the role of microbe in N-containing gas formations and diffusions, mineralization/immobilization, nitrification/denitrification, etc. The figure 1 needs to include such information on microbial processes. 4. The description of major equations is barely understandable for readers. There is no connection between these equations listed in the manuscript. The authors should provide equations focusing on N2O fluxes. 5. The authors mentioned their previous model and used it to make comparison with the current version. They should have a description of their previous model and list the improvements in the methodology. 6. They emphasized site-level estimates and climate data sources, but not for global simulations. There is no detailed information on the climate data source or description on climate variability during 1990-2000. 7. The authors should provide the method on how to extrapolate site-level estimates to the global level. Also, I am curious with the uncertainty range (7.1-10.3 Tg N yr-1), but they did not give any explanations. 8. As also indicated in the manuscript, biological N fixation and denitrification can contribute a significant amount of N2O emissions, but these processes were not included in this study. A paragraph should be included in the discussion sector to address this ignorance and its impact on the entire estimates. 9. They claimed that CN ratio plays a significant role in N2O emissions, which is one of their objectives. They indeed mentioned CN ratio threshold in the methodology; however, nothing

special has been described in the result or discussion sectors. 10. The improved trait-based model is actually a hybrid of first-order and second-order expression. According to Fig. 4, I cannot tell the advantages of this improved model. They should provide more evidence. 11. They found that tropical peatland has the highest N2O emission, up to 5.7 kg N ha-1 yr-1. When I go back to that article, they chose this site because the peatland was converted to cropland and induced a much higher N2O emissions. However, I guess this model is incapable to simulate land conversion and its impact. If you used this site, your estimates in Southeast Asia should be much higher than other previous studies. Thus, it is not appropriate using this site for model calibration. 12. In Fig. 2, I can only detect one site in the Congo Basin for model calibration. Based on my knowledge, this region may be a large source for N2O emission. Thus, I suggest the authors to collect more data to re-calibrate their model. 13. The microbial biomass data was not well explained. We need to see more details about these data. 14. It seems that the tables and figures can be further improved. For example, the table 1 and table 2 can be provided as a supplement file. The Fig.2 can be improved by removing the Antarctica regions. Fig. 3, 4 and 6 should be improved as the current resolution of figures is poor. In addition, there are two fig.4. 15. Table 3a conflicted with details provided in section 2.3 of the paper, and tables 3 and 4 should be swapped to match the order given in the methods section. There was also repetition within the methods sections. 16. Literature cited: Several new efforts in soil N2O modeling have been published recently. Literature review should include recent modeling efforts. Particularly, I am surprised that the authors did not recognize a major NO2 model intercomparison project- NMIP (Tian et al 2018). Tian, H., J. Yang, C. Lu, R. Xu, J. G Canadell, R. B. Jackson, et al. (2018) The global N2O Model Intercomparison Project, Bulletin of the American Meteorological Society (BAMS), https://doi.org/10.1175/BAMS-D-17-0212.1

---

## Short Comment (SC1) · 5 Nov 2018

In line 15 on page 6, the authors wrote that "In addition, the processes of N deposition, mineralization, and denitrification were also modeled." The detailed modeling approaches for these processes might have been included in the previous publication of the TEM model already, but I think the authors perhaps should present them at least briefly here. Especially, I cannot find how the authors obtain the N deposition data. And how it is incoporated in the models.

---

## Referee Comment (RC2) · Anonymous Referee #2 · 6 Nov 2018

Yu and Zhuang improved the N2O emission processes in one existing land ecosystem model by using trait-based biogeochemistry models. Trait-based modeling is a new direction for model development. This could potentially improve model. However, I think this paper has some deficits and drawbacks need to be addressed.

1. The authors modified model nitrification process. As I know, most of soil N2O emission is from denitrification process, in which NO3- is converted to N2, N2O, and NO. Only a small part of N2O is from the nitrification process. I don't think the improvement in nitrification process could substantially improve the simulated N2O. I would suggest the authors use trait-based approach to represent denitrification as well.

[Figure]

2. The equations in original TEM should be described.

3. The authors claim the nitrification process was improved. However, nitrification rate was not validated.

4. For model sensitivity, authors examined model sensitivity to climate and soil C/N. It is correct that N2O emission is sensitive to climate change (particularly temperature). However, N2O emissions in the natural ecosystems could be very sensitive to the atmospheric N deposition. In recent years, there is a debate on how soil N2O emissions response to CO2 concentration. I would see some results about N2O sensitivity to N deposition and CO2.

5. what is the date sources of atmospheric CO2 and nitrogen deposition?

6. Recently, a global N2O model comparison has been initiated to run models from 1860 to 2016 (Tian et al., 2018). Ten land models were included in this project. The participating models include both natural system and cropland soils. I would suggest the authors to justify why this paper only included natural soils but ignored the more important N fertilizer in cropland.

---

## Author Comment (AC1) · 11 Dec 2018

Response: Thank you for the overall positive feedback. We have thoroughly revised the paper following your comments and suggestions.

1. The authors intent to estimate N2O emission from natural soils during 1990-2000. However, the terrestrial ecosystems have been extensively disturbed and managed. It's unclear about what the natural soils mean in this manuscript. There is no detail information on how they generate the natural ecosystem data across the global land surface. Response: In this study, we did not include the processes introduced by human activities. A natural ecosystem here contains matured and undisturbed soil

and vegetation excluding the effect of land use. For natural ecosystems data we used, please refer to Section 2.2 about the data organization. Specifically, the land cover data include soil texture and vegetation type, which are inherited from Melillo et al. (1993) and Zhuang et al. (2003).

2. Global map (Fig. 5) shows N2O emissions from the cultivated areas where crops planted during 1990-2000 according to my knowledge. Do you consider background emission from cropland as natural emissions? Or you treat cropland as other types of vegetation? Response: In this study, we only considered natural ecosystem emissions. Croplands emission were not modeled. Please refer to Fig. 2 (top) from Melillo et al. (1993) for the distribution of natural ecosystems.

3. It needs to provide more explicit explanation on the role of microbe in N-containing gas formations and diffusions, mineralization/immobilization, nitrification/denitrification, etc. The figure 1 needs to include such information on microbial processes. Response: We revised Figure 1 to delete N fixation process that was not considered in this study. Now the figure 1 caption is "Schematic diagram of N2O emissions and N cycling between plants, soils, and the atmosphere: The input of N from the atmosphere to soils through nitrogen deposition as nitrate and ammonia; microbial biomass dynamics were modeled; Nitrification is modeled as a function of microbial biomass, soil organic nitrogen, and physical conditions, more details refer to Yu (2016); N uptake by plants is modeled in original TEM (McGuire et al., 1992)."

4. The description of major equations is barely understandable for readers. There is no connection between these equations listed in the manuscript. The authors should provide equations focusing on N2O fluxes. Response: The major equations listed in the text describe the trait of microbes related to nitrification. Limited by the length of the manuscript, more equations focusing on N2O fluxes and other processes of N cycle can be referred to in Master's thesis of Yu, T. (2016). Tong Yu (2016), Quantifying the global N2O emissions from natural ecosystems using a mechanistically-based biogeochemistry model, MS thesis, http://docs.lib.purdue.edu/dissertations/AAI10145857/

5. The authors mentioned their previous model and used it to make comparison with the current version. They should have a description of their previous model and list the improvements in the methodology. Response: Thank you. We now added a brief description of our previous model in 2.2 Model Modification section. The improvements are described from line 20 to line 25 of Page 6.

6. They emphasized site-level estimates and climate data sources, but not for global simulations. There is no detailed information on the climate data source or description on climate variability during 1990-2000. Response: Thanks for pointing this out. For the global simulation period, the monthly air temperature, precipitation, cloudiness and vapor pressure data are from Climate Research Unit (CRU).

7. The authors should provide the method on how to extrapolate site-level estimates to the global level. Also, I am curious with the uncertainty range (7.1-10.3 Tg N yr-1), but they did not give any explanations. Response: Please refer to Section 2.2 about the global forcing data organization and Section 2.3 for the parameters used for global simulation. We added a couple of sentences to describe how the parameters obtained from site-level parameterization to the global scale "We apply the site-level parameters for representative ecosystem types to grid cells at 0.5o x 0.5o resolution at the global scale. The ecosystem types are listed in Table 2 and their distributions are from Melillo et al. (1993)."

The uncertainty range of simulated N2O emissions is induced from the range of parameters shown in Table 1. We also made this clear in the text line 10-12 on page 13 by adding a sentence "The uncertainty range of simulated N2O emissions is induced from the range of parameters shown in Table 1" (section 3.3).

8. As also indicated in the manuscript, biological N fixation and denitrification can contribute a significant amount of N2O emissions, but these processes were not included in this study. A paragraph should be included in the discussion sector to address this ignorance and its impact on the entire estimates. Response: Denitrification is simulated in this study, which can be referred to Yu (2016). Due to the length limitation, we want to refer the detailed description of denitrification process to Yu (2016). N fixation is not considered in this study. We deleted that process in Figure 1. In addition, we now added a paragraph in Section 4.3 to discuss the microbial effects on denitrification and effect of N fixation on current estimation.

9. They claimed that CN ratio plays a significant role in N2O emissions, which is one of their objectives. They indeed mentioned CN ratio threshold in the methodology; however, nothing special has been described in the result or discussion sectors. Response: In this revision, in Result and Discussion sections, we did sensitivity tests on the effects of soil carbon and soil nitrogen, the latter is highly related to the growth and metabolism of microbes, in turn, affecting N2O emissions.

10. The improved trait-based model is actually a hybrid of first-order and second-order expression. According to Fig. 4, I cannot tell the advantages of this improved model. They should provide more evidence. Response: Figure 4 showed some improvement based on both slope and R2 values from two versions of model simulations. The previous model had a comparatively smaller R2 and slope in comparison with observations, but overestimated N2O emissions because the model ignored the N taken up by soil microbes. The comparison is shown in Section 3.1, page 9-10. Figure 5 showed that the improved trait-based model has a better performance by capturing seasonal variations. The comparison is shown in Section 3.1, page 10.

11. They found that tropical peatland has the highest N2O emission, up to 5.7 kg N ha-1 yr-1. When I go back to that article, they chose this site because the peatland was converted to cropland and induced a much higher N2O emissions. However, I guess this model is incapable to simulate land conversion and its impact. If you used this site, your estimates in Southeast Asia should be much higher than other previous studies. Thus, it is not appropriate using this site for model calibration. Response: Due to scarcity of quality observational data, we made a compromise to use this site data to do calibration. In this revision, we pointed that out in the text line 19 – 21 on page 10 by

adding "We recognized the site data used from a cropland ecosystem converted from peatlands, which may be with higher N2O emissions than natural ecosystems in the region. This may result in relatively high emissions from this type of land ecosystems in the region".

12. In Fig. 2, I can only detect one site in the Congo Basin for model calibration. Based on my knowledge, this region may be a large source for N2O emission. Thus, I suggest the authors to collect more data to re-calibrate their model. Response: Thanks for the suggestion. We have not been able to get more observed data for this region. We will keep eye on the literature to obtain more data when the data are available to improve our simulations in future studies.

13. The microbial biomass data was not well explained. We need to see more details about these data. Response: In this revision, we added a brief introduction to the source and organization of microbial biomass data from line 3 to 5 of Page 8. Limited by the length of our text, please refer to the following paper for more details about the original dataset:

Xu, X., P.E. Thornton, and P. Potapov. 2014. A Compilation of Global Soil Microbial Biomass Carbon, Nitrogen, and Phosphorus Data. ORNL DAAC, Oak Ridge, Tennessee, USA. https://doi.org/10.3334/ORNLDAAC/1264

14. It seems that the tables and figures can be further improved. For example, the table 1 and table 2 can be provided as a supplement file. The Fig.2 can be improved by removing the Antarctica regions. Fig. 3, 4 and 6 should be improved as the current resolution of figures is poor. In addition, there are two fig.4. Response: Thank you for your suggestion. We have fixed the error on Figure 4. We have removed the Antarctic region in figure 2. The resolution of figures was adjusted according to demand of journal.

15. Table 3a conflicted with details provided in section 2.3 of the paper, and tables 3 and 4 should be swapped to match the order given in the methods section. There

was also repetition within the methods sections. Response: Thank you for your careful review. We have corrected the error.

16. Literature cited: Several new efforts in soil N2O modeling have been published recently. Literature review should include recent modeling efforts. Particularly, I am surprised that the authors did not recognize a major NO2 model inter comparison project- NMIP (Tian et al 2018). Tian, H., J. Yang, C. Lu, R. Xu, J. G Canadell, R. B.Jackson, et al. (2018) The global N2O Model Intercomparison Project, Bulletin of the American Meteorological Society (BAMS), https://doi.org/10.1175/BAMS-D-17-0212.1 Response: Thank you for the reference. We have carefully read this paper and added related results to Discussion section on page 14-15.

---

## Author Comment (AC2) · 11 Dec 2018

In line 15 on page 6, the authors wrote that "In addition, the processes of N deposition, mineralization, and denitrification were also modeled." The detailed modeling approaches for these processes might have been included in the previous publication of the TEM model already, but I think the authors perhaps should present them at least briefly here. Especially, I cannot find how the authors obtain the N deposition data. And how it is incorporated in the models.

Response: Thank you much for your suggestions. Limited by the length of the manuscript, more equations focusing on N2O fluxes and other processes of N cy-

cle can be referred to a Master thesis of Yu, T. (2016). Tong Yu (2016), Quantifying the global N2O emissions from natural ecosystems using a mechanistically-based biogeochemistry model, MS thesis, http://docs.lib.purdue.edu/dissertations/AAI10145857/ For the second comment about N deposition data, we added a paragraph in Section 2.2 Data. The N deposit data was directly applied as part of input of ammonia and nitrate.

---

## Author Comment (AC3) · 11 Dec 2018

1. The authors modified model nitrification process. As I know, most of soil N2O emission is from denitrification process, in which NO3- is converted to N2, N2O, and NO. Only a small part of N2O is from the nitrification process. I don't think the improvement in nitrification process could substantially improve the simulated N2O. I would suggest the authors use trait-based approach to represent denitrification as well. Response: Thank you for your suggestions. Denitrification is definitely an import process as it contributes more especially in reduced environment. In this revision, we added a paragraph about the potential effect of denitrification in Discussion section 4.3 (line 13-20,

page 17).

2. The equations in original TEM should be described. Response: Limited by the length of the manuscript, more equations focusing on N2O fluxes and other processes of N cycle can be referred to a Master thesis of Yu, T. (2016). Tong Yu (2016), Quantifying the global N2O emissions from natural ecosystems using a mechanistically-based biogeochemistry model, MS thesis, http://docs.lib.purdue.edu/dissertations/AAI10145857/

3. The authors claim the nitrification process was improved. However, nitrification rate was not validated. Response: Because direct observational data for nitrification rate is too few to allow us conduct its validation. Instead, we validated modeled N2O emissions by comparing with observed data.

4. For model sensitivity, authors examined model sensitivity to climate and soil C/N. It is correct that N2O emission is sensitive to climate change (particularly temperature). However, N2O emissions in the natural ecosystems could be very sensitive to the atmospheric N deposition. In recent years, there is a debate on how soil N2O emissions response to CO2 concentration. I would see some results about N2O sensitivity to N deposition and CO2. Response: In this revision, we conducted the sensitivity test on the effects of dry and wet N deposition on N2O emissions, and added it to Section 2.3 and 3.2.1. The average atmospheric CO2 was applied uniformly for each grid, so we did not do the sensitivity test on CO2 effects. In our future work, we will obtain spatially and temporally explicit CO2 data to drive the model to examine the CO2 effects on N2O emissions. This step will take a significant effort, which is beyond this study.

5. What is the date sources of atmospheric CO2 and nitrogen deposition? Response: In this revision, we added the data sources in Section 2.2 Data.

6. Recently, a global N2O model comparison has been initiated to run models from 1860 to 2016 (Tian et al., 2018). Ten land models were included in this project. The participating models include both natural system and cropland soils. I would suggest the authors to justify why this paper only included natural soils but ignored the more

important N fertilizer in cropland. Response: Thank you for the reference. We have carefully read this paper and added related results to Discussion on page 14.

---

## Author Comment (AC4) · 11 Dec 2018

Attached is my revised manuscript based on your comments.

Please also note the supplement to this comment:
https://www.biogeosciences-discuss.net/bg-2018-377/bg-2018-377-AC4-supplement.pdf

---

## Author Comment (AC5) · 11 Dec 2018

[revised manuscript text omitted]

(2)

Where  $[NO_2]$  represents the concentration of NO2.  $V_{Ox}^{NO}$  is the oxidization rate by NOB and  $D_{AOO}^{NO_2}$  is the loss in the detoxification.

The consumption rate of NH3 by AOA and AOB is determined by the concentration of NH3 and O2 in the soil. For the simulation of ammonia oxidation by ammonia-oxidizing organism, the cell biomass was considered in the Briggs-Haldane kinetics calculation (Koper et al., 2010):

$$V_{AOO}^{NH_3} = V_{max}^{NH_3} \frac{[NH_3]}{\kappa_{AOO}^{NH_3} + [NH_3] \left(\frac{1 + [NH_3]}{\kappa_{AOO}^{NH_3}}\right)} \frac{[O_2]}{\kappa_M^{O_2} + [O_2]} B_{TA}$$
(3)

Where  $V_{max}^{NH_3}$  is the maximum substrate uptake rate for ammonia (M day-1). This value varies between different guilds of microbes. The average value for AOB is about 0.5 and the average value for AOA is about 0.6.  $K_{AOO}^{NH_3}$  is the half saturation constant for NH3 (µM) and  $K_M^{O_2}$  is the Michaelis-Menten parameter for oxygen (µM) (Table 1).  $B_{TA}$ is the total cell biomass for ammonia oxidizing organisms (AOA+AOB).

The consumption of  $NO_2^-$  is similar to Eq.3:

$$V_{NOB}^{NO_2} = V_{max}^{NO_2} \frac{[NO_2]}{K_M^{NO_2} + [NO_2]} \frac{[O_2]}{K_M^{O_2} + [O_2]} B_{TN}$$
(4)

20

15

Where,  $K_M^{NO_2}$  is the maximum substrate uptake rate for NO2-(M day-1). This value also depends on different guilds, and the value could be from 0.4 to 4 (Bouskill et al., 2012); here 2.0 was used.  $K_M^{NO_2}$  is the half saturation constant for NH3 (µM) and  $K_M^{O_2}$  is the Michaelis-Menten parameter for oxygen (µM).  $B_{TN}$  represents the total cell biomass of NOB.

Considering the cell division of microbes, the growth of AOB biomass is (Bouskill et al., 2012):

25
$$\frac{dB_{TA}}{dt} = \mu_{max} min\{d_i\} B_{TA} - \varepsilon B_{TA} - \frac{1}{4} \left( D_A^{NO_2} + D_A^{NO} \right)$$
(5)

The first term  $\mu_{max}min\{d_i\}B_{TA}$  is the cell division rate.  $\mu_{max}$  (day-1) is the nitrifier maximum specific growth rate for ammonia oxidizing organisms (AOO). It is less than 0.1 for AOO, and here 0.05 was used.  $min\{d_i\}$  represents the constraint of element. It is defined as the cell division of AOO or NOB, which is governed by Droop kinetics (Droop, 1973):

$$\qquad d_B^i = max \left( 1 - \frac{q_B^{min}}{q_B^i}, 0 \right) \tag{6}$$

Q is the cellular quota for nitrogen or carbon. It is defined as  $Q_N = B_N/B_T$ ,  $Q_C = B_C/B_T$ , which is the percentage of a certain element in total biomass. For example, the cell division of N for a guild is:

$$d_{B,N}^{1} = \max\left(1 - \frac{1/13.2}{B_{N}/(B_{N} + B_{C})}, 0\right)$$
(7)

According to the C: N ratio for nitrifiers, the amount of carbon is supposed to be 6.6 to 13.2 times of the amount of
 N (Bouskill et al., 2012). If the ratio of C: N is greater than 1/13.2, the reproduction of microbe is limited by N. In contrast, the process is limited by C if C: N is smaller than 6.6.

The second term  $\varepsilon B_{TA}$  indicates the death rate.  $\varepsilon$  is the mortality rate. The last term  $\frac{1}{4} (D_A^{NO_2} + D_A^{NO})$  refers to the biomass loss for converting NO2 to NO and NO to N2O:

$$4NO_2 + CH_2O \to 4NO + CO_2 + 3H_2O$$
15
$$8NO + 2CH_2O \to 4N_2O + 2CO_2 + 2H_2O$$
(8)

Similarly, the growth of NOB biomass is (Bouskill et al., 2012):

[revised manuscript text omitted]

Savanna            | 145.5     | -17.5      | 19.0-24.3           | 69.7-236.1
104.8(dry | 10~18                | 0.07~0.20                                                           | Breuer et al.(2000)                  |
| Chagurarama, Guarico State, Venezuela                 | (grassland)
Savanna           | -79.5     | 36.5       | 3.5                 | season)                 | 9                    | 0.01                                                                | Hao et al.(1988)                     |
| 10km from No 4
Lake Creek, Linn County Williamette | (woodland)                       | -79.5     | 36.5       | 3.5                 | 104.8                   | 9                    | 0.03                                                                | Hao et al.(1988)
Horwath et al    |
| Valley, Oregon                                        | Grass                            | -123.5    | 44.5       | 10.7                | 305.7                   | 93                   | 0.31                                                                | (1998)
Butterbach-Bahl et         |
| Höglwald, Germany                                     | Coniferous                       | 14        | 51         | 14.6                | 66.8                    | 30                   | 0.04~0.12                                                           | al(1997)                             |
| Kiel, Germany                                         | Deciduous                        | 112.5     | 23         | 21.4                | 1927                    | 365                  | 0.4~4.9                                                             | Mogge et al.(1998)
Seiler and     |
| Mainz, Germany                                        | Grass                            | 8.5       | 50         | 10                  | 45.5-546                | 32-71                | 0.02-0.13                                                           | Conrad(1981)
Butterbach-Bahl et   |
| Ballyhooly, Republic of Ireland                       | Coniferous                       | -8.5      | 52         | 9.6                 | 89.9                    | 3                    | 0                                                                   | al(1998)
Goossens et              |
| Poppel, Belgium                                       | Deciduous                        | 5         | 51.5       | 11                  | 657-1017.6              | 317-365              | 0                                                                   | al.(2001)
Pitcaim et al.          |
| Central Scotland
Guanica Commonwealth Forest, SW   | Deciduous
Tropical Dry        | -4.5      | 56.5       | 8.7                 | 828.8
108.4-         | 210                  | 1.15~2.29                                                           | (2002)
Erickson et al             |
| Puerto Rico                                           | forest
Mediterranean          | -63       | -10        | 25.6-26.3           | 1626.4                  | 153-365              | 0.02-0.7                                                            | (2002)
Anderson and               |
| San Dimas Experiment Forest                           | Shrub lands                      | -118      | 34         | 13-42               | 696                     | 60                   | 0.05~0.15                                                           | Poth (1989)
Müller and Shelock    |
| Lincolen Canterbury, New Zealand                      | Grassland                        | 172.5     | -43.5      | 1~21                | 2-47                    | 400                  | 0.255                                                               | (2004)                               |
| Nylsvley Nature Reserve, South Africa                 | Savanna
Tropical              | -24.5     | 28.5       | 12~15               | 625                     | 19                   | 0.28                                                                | Scholes et al.(1997)                 |
| Gambutt, South Kalimantan, Indonesia                  | Peatland                         | 114.5     | 3.5        | 28                  |                         | 28                   | 5.698                                                               | Hadi et al. (2000)                   |
| Barambai, South Kalimantan, Indonesia                 | Tropical forest
Tropical rain | 114.5     | 3          | 28                  |                         | 28                   | 2.628                                                               | Hadi et al. (2000)
Davidson et al |
| Fazenda Victória, Brazil                              | forest                           | 55        | 3          |                     | 1800                    | 1.5 yr               | 2.6                                                                 | (2004)                               |
| Orinoco Ilanos, Venezuela                             | Savanna                          | -63.5     | 9.5 | 27.3                | 992                     | 1 yr                 | 0.73                                                                | Simona et al (2004)                  |

| Horquetas, Costa Rica                                                     | Tropical Pastures                     | -85    | 10   | 25.8                  | 3962      | 23-30     | 2.365     | Veldkamp et al.
(1998)
Keller& Reiners |
|---------------------------------------------------------------------------|---------------------------------------|--------|------|-----------------------|-----------|-----------|-----------|----------------------------------------------|
| La Selva biological station, Costa Rica                                   | Tropical Forest                       | -84    | 10.5 | 25.8                  | 3962      | 15months  | 3.74      | (1994)
Mosjer& Delgado                    |
| Isabela and Mayaguez, Puerto Rico
Luquillo Experimental Forest, Puerto | Tropical Grassland
Subtropical wet | -67    | 18   |                       |           | 1 yr      | 1.51      | (1997),
Mosier(1997a)
Erickson et al   |
| Rico                                                                      | Forest                                | -66    | 18   | 23.5-27               | 2900-3200 | 1 yr      | 1.75      | (2001)                                       |
| Kilauea, Hawaii                                                           | Rain forest                           | -155.5 | 19.5 |                       |           | 1 yr      | 0.223     | Riley & Vitousek
(1995)                   |
| Wudaoliang, Qinghai, China                                                | Alpine Grassland                      | 93     | 35   | -5.6
(-10.5)       | 200-400   | 1yr       | 0.069     | Pei (2003)                                   |
| Mount Taylor, New Mexico                                                  | Temperate forest                      | -107.5 | 35.5 | to 17                 | 640-720   | 2yrs      | 0.03 0.09 | Matson et al (1992)                          |
| Nevada Desert FACE facility, US                                           | Desert                                | -116   | 37   |                       | 140       | 2yrs      | 0.11      | Billings et al.(2002)
Mosier& Delgado     |
| Colorado, USA                                                             | Temperate
Grassland                | -104.5 | 40.5 | (-5) to 30
(-7, 3) | 350       | 3yrs      | 0.167     | (1997),
Mosier(1997a)                     |
| Changbai Mountain Forest Research Star                                    | tion                                  | 127    | 41.5 | (=7.5)
to 3.3      | 700-1300  | 138       | 0.28      | Chen et al (2000)                            |
| USA                                                                       | SageBrush Steppe                      | -107   | 41.5 | 2.7
(-8) to        | 525       | 2 yrs     | 0.21      | Matson et al (1991)                          |
| Harvard Forest, USA                                                       | Temperate Forest                      | -72    | 42.5 | 23                    |           | 2yrs      | 0.02 0.06 | Bowden et al(1990)                           |
| Whiteface Mt, NY, USA                                                     | Temperate Forest                      | -74    | 44.5 | 0~18                  |           | 1990      | 0.185     | Castro et al (1992)                          |
| Mt Mansfield, VT, USA                                                     | Temperate Forest                      | -73    | 44.5 | 4~19                  |           | 1990      | 0.1708    | Castro et al (1992)                          |
| Mt Ascutney, VT, USA                                                      | Temperate Forest                      | -72.5  | 43   | 6~22                  |           | 1990      | -0.098    | Castro et al (1992)                          |
| Mt Washington, NH, USA                                                    | Temperate Forest                      | -71    | 44   | 4~17                  |           | 1990      | -0.02     | Castro et al (1992)                          |
| Acadia, ME, USA                                                           | Temperate Forest                      | -68.5  | 44   | 5~20                  |           | 1990      | 0.0315    | Castro et al (1992)                          |
| Waldhausen, Germany                                                       | Temperate Forest                      | 10     | 49   | 0.4)~20               |           | 1981-1982 | 0.473     | Shmidt et al. (1988)                         |
| Bechenheim, Germany                                                       | Temperate Forest                      | 8      | 49.5 | 0~18.3                |           | 1981-1982 | 0.802     | Shmidt et al. (1988)                         |
| Langenlonsheim, Germany                                                   | Temperate Forest                      | 8      | 50   | 0~18.6                |           | 1981-1982 | 0.714     | Shmidt et al. (1988)                         |

Table 3: Sensitivity Studies of  $N_2O$  emissions (%) responding to changes of: (a) climate and soil data at different levels; (b) temperature at 5% and 20% for different vegetation types; (c) precipitation at 5% and 20% for different vegetation types

|                            | 5%    | -5%    | 10%  | -10%   | 20%  | -20%  |
|----------------------------|-------|--------|------|--------|------|-------|
| Air temperature            | 3.2   | -2.5   | 1.2  | -5.5   | -11  | -17   |
| Precipitation              | 4.5   | -1.8   | 0.97 | -3.4   | -6   | -10   |
| Cloudiness                 | -0.85 | 0.43   | -3.2 | 1.1    | -5   | 0.9   |
| Water Vapor Pressure       | 0.03  | -0.015 | 0.07 | -0.032 | 0.1  | -0.92 |
| Soil Carbon                | 0.8   | -0.7   | 1.5  | -1.6   | 2.9  | -3.2  |
| Soil Nitrogen              | 0.2   | -0.17  | 0.24 | -0.25  | 0.27 | -0.3  |
| Dry Deposit N              | 0.18  | -0.23  | 0.65 | -0.60  | 3.5  | -2.4  |
| Wet Deposit N - | 7.2   | -8.5   | 18   | -17    | 33   | -29   |

(a)

(b)

|                             | 5%  | -5%  | 20% | -20% |
|-----------------------------|-----|------|-----|------|
| Tropical Forest             | -1  | -0.5 | -19 | -11  |
| Temperate Evergreen Forest  | 6.5 | -4   | -6  | -13  |
| Temperate Deciduous Forest  | 4.3 | -5.5 | -7  | -15  |
| Temperate Coniferous Forest | 8.6 | -4.2 | 3   | -37  |
| Temperate Grassland         | 2.1 | -3.5 | -11 | -19  |
| Savanna                     | 0.5 | -2   | -16 | -7.2 |
| Succulent                   | -2  | -0.2 | -24 | -5.5 |
| Mediterranean Shrub lands   | 0.7 | -1.5 | -17 | -12  |
| Tundra                      | 5.5 | -6.2 | 3.5 | -27  |
|                             |     |      |     |      |

|                             | 5%  | -5%  | 20%  | -20% |
|-----------------------------|-----|------|------|------|
| Tropical Forest             | 0.7 | -0.3 | -11  | -12  |
| Temperate Evergreen Forest  | 2.6 | -3.5 | -8.2 | -12  |
| Temperate Deciduous Forest  | 4.2 | -0.8 | -9   | -8   |
| Temperate Coniferous Forest | 1.5 | -2.2 | -5.3 | -9.7 |
| Temperate Grassland         | 4.6 | -3.3 | -2.6 | -12  |
| Savanna                     | 5.7 | -2.8 | -5.3 | -17  |
| Succulent                   | 4.4 | -6.3 | -2.7 | -18  |
| Mediterranean Shrub lands   | 2.2 | -3.7 | -6.5 | -15  |
| Tundra                      | 0.2 | -0.2 | -3.1 | -11  |
|                             |     |      |      |      |

Table 4: Key parameters' values after calibration

|                             | Vmax_AOO  | Vmax_NOB  | miu_max | K_NH | K_NO | K_O  |
|-----------------------------|-----------|-----------|---------|------|------|------|
|                             | (M day-1) | (M day-1) | (day-1) | (μΜ) | (µM) | (µM) |
| Tropical Forest             | 0.54      | 3.5       | 0.06    | 56   | 100  | 6.8  |
| Temperate Evergreen Forest  | 0.52      | 3         | 0.05    | 46   | 90   | 7.2  |
| Temperate Deciduous Forest  | 0.5       | 3         | 0.05    | 48   | 88   | 7    |
| Temperate Coniferous Forest | 0.52      | 3.2       | 0.05    | 46   | 82   | 7    |
| Temperate Grassland         | 0.5       | 2.5       | 0.05    | 38   | 60   | 12   |
| Savanna                     | 0.5       | 2.5       | 0.04    | 42   | 62   | 12   |
| Succulent                   | 0.46      | 1         | 0.04    | 22   | 52   | 14   |
| Medeterranean Shrub lands   | 0.48      | 2         | 0.04    | 40   | 66   | 14   |
| Tundra                      | 0.48      | 2.5       | 0.05    | 40   | 68   | 4.2  |

| Table 5: Sensitivity | (%) of key | parameters | for biomes |
|----------------------|------------|------------|------------|
|----------------------|------------|------------|------------|

|                                 | 5%    | -5%  | 25%   | -25% |
|---------------------------------|-------|------|-------|------|
| Vmax_AOO (M day -1 ) | 1.3   | -3.1 | 7     | -9.9 |
| Vmax_NOB (M day-1)              | 0.8   | -2   | 5.5   | -7.5 |
| miu_max (day -1 )    | 2.2   | -1.3 | 8.7   | -9.7 |
| K_NH (μM)                       | -0.25 | 0.26 | -0.52 | 0.38 |
| Κ_ΝΟ (μΜ)                       | -0.15 | 0.28 | -0.17 | 0.3  |
| Κ_Ο (μΜ)                        | -0.23 | 0.24 | -0.14 | 2    |

Figure 1: Schematic diagram of N2O emissions and N cycling between plants, soils, and the atmosphere: The input of N from the atmosphere to soils through nitrogen deposition as nitrate and ammonia; microbial biomass dynamics were modeled; Nitrification is modeled as a function of microbial biomass, soil organic nitrogen, and physical conditions, more details refer to Yu (2016); N uptake by plants is modeled in original TEM (McGuire et al., 1992).

---

## Author Comment (AC6) · 11 Dec 2018

Attached is my revised manuscript based on your comments.

Please also note the supplement to this comment:
https://www.biogeosciences-discuss.net/bg-2018-377/bg-2018-377-AC6-supplement.pdf